# Generalized Linear Bandits with Limited Adaptivity

**Ayush Sawarni**[*]
Stanford University
ayushsaw@stanford.edu

**Nirjhar Das**[†]
Indian Institute of Science Bangalore
nirjhardas@iisc.ac.in

**Siddharth Barman**
Indian Institute of Science Bangalore
barman@iisc.ac.in

**Gaurav Sinha**
Microsoft Research India
gauravsinha@microsoft.com

## Abstract

We study the generalized linear contextual bandit problem within the constraints of limited adaptivity. In this paper, we present two algorithms, B-GLinCB and RS-GLinCB, that address, respectively, two prevalent limited adaptivity settings. Given a budget $M$ on the number of policy updates, in the first setting, the algorithm needs to decide upfront $M$ rounds at which it will update its policy, while in the second setting it can adaptively perform $M$ policy updates during its course. For the first setting, we design an algorithm B-GLinCB, that incurs $\tilde{O}(\sqrt{T})$ regret when $M = \Omega\left(\log \log T\right)$ and the arm feature vectors are generated stochastically. For the second setting, we design an algorithm RS-GLinCB that updates its policy $\tilde{O}(\log^2 T)$ times and achieves a regret of $\tilde{O}(\sqrt{T})$ even when the arm feature vectors are adversarially generated. Notably, in these bounds, we manage to eliminate the dependence on a key instance dependent parameter $\kappa$, that captures non-linearity of the underlying reward model. Our novel approach for removing this dependence for generalized linear contextual bandits might be of independent interest.

## 1 Introduction

Contextual Bandits (CB) is an archetypal framework that models sequential decision making in time-varying environments. In this framework, the algorithm (decision maker) is presented, in each round, with a set of arms (represented as $d$-dimensional feature vectors), and it needs to decide which arm to play. Once an arm is played, a reward corresponding to the played arm is accrued. The regret of the round is defined as the difference between the maximum reward possible in that round and the reward of the played arm. The goal is to design a policy for selecting arms that minimizes cumulative regret (referred to as the regret of the algorithm) over a specified number of rounds, $T$. In the last few decades, much progress has been made in designing algorithms for special classes of reward models, e.g. linear model [3, 4, 1, 16], logistic model [6, 2, 7, 28] and generalized linear models [8, 19].

However, despite this progress, there is a key challenge that prevents deployment of CB algorithms in the real world. Practical situations often allow for very limited adaptivity, i.e., do not allow CB algorithms to update their policy at all rounds. For example, in clinical trials [10], each trial involves administering medical treatments to a cohort of patients, with medical outcomes observed and collected for the entire cohort at the conclusion of the trial. This data is then used to design the treatment for the next phase of the trial. Similarly, in online advertising [25] and recommendations [18], updating the policy after every iteration during deployment is often infeasible due to infrastructural constraints. A recent line of work [23, 22, 11, 12, 21, 9, 26] tries to address this limitation

---

[*]Work done while author was at Microsoft Research India

[†]Work done while author was at Microsoft Research India

38th Conference on Neural Information Processing Systems (NeurIPS 2024).

by developing algorithms that try to minimize cumulative regret while ensuring that only a limited number of policy updates occur. Across these works, two settings (called **M1** and **M2** from here onwards) of limited adaptivity have been popular. Both **M1, M2** provide a budget $M$ to the algorithm, determining the number of times it can update its policy. In **M1** [23, 13], the algorithm is required to decide upfront a sub-sequence of $M$ rounds where policy updates will occur. While in **M2** [1, 23]), the algorithm is allowed to adaptively decide (during its course) when to update its policy.

Limited adaptivity algorithms were recently proposed for the CB problem with linear reward models under the **M1** setting [23, 11], and optimal regret guarantees were obtained when the arm feature vectors were stochastically generated. Similarly, in their seminal work on linear bandits, [1] developed algorithms for the **M2** setting and proved optimal regret guarantees with no restrictions on the arm vectors. While these results provide tight regret guarantees for linear reward models, extending them to generalized linear models is quite a challenge. Straightforward extensions lead to sub-optimal regret with a significantly worse dependence on an instance dependent parameter $\kappa$ (See Section 2 for definition) that captures non-linearity of the problem instance. In fact, to the best of our knowledge, developing optimal algorithms for the CB problem with generalized linear reward models under the limited adaptivity settings **M1**, **M2**, is an open research question. This is the main focus of our work. We make the following contributions.

## 1.1 Our Contributions

• We propose B-GLinCB, an algorithm that solves the CB problem for bounded (almost surely) generalized linear reward models (Definition 2.1) under the **M1** setting of limited adaptivity. We prove that, when the arm feature vectors are generated stochastically, the regret of B-GLinCB at the end of $T$ rounds is $\tilde{O}(\sqrt{T})$, when $M = \Omega(\log \log T)$. When $M = O(\log \log T)$, we prove an $\tilde{O}(T^{2^{M-1}/(2^M-2)})$ regret guarantee. While the algorithm bears a slight resemblance to the one in [23], direct utilization of their key techniques (distributional optimal design) results in a regret guarantee that scales linearly with the instance dependent non-linearity $\kappa$. On the other hand, the leading terms in our regret guarantee for B-GLinCB have no dependence on $\kappa$. To achieve this, we make novel modifications to the key technique of distributional optimal design in [23]. Along with this, the rounds for policy updates are also chosen more carefully (in a $\kappa$ dependent fashion), leading to a stronger regret guarantee.

• We propose RS-GLinCB, an algorithm that solves the CB problem for bounded (almost surely) generalized linear reward models (Definition 2.1) under the **M2** setting of limited adaptivity. RS-GLinCB builds on a similar algorithm in [1] by adding a novel context-dependent criterion for determining if a policy update is needed. This new criterion allows us to prove optimal regret guarantee ($\tilde{O}(\sqrt{T})$) with only $O(\log^2 T)$ updates to the policy. It is quite crucial for the generalized linear reward settings since, without it, the resultant regret guarantees have a linear dependence on $\kappa$.

• Our work also resolves a conjecture in [17] by proving an optimal ($\tilde{O}(\sqrt{T})$) regret guarantee (for the CB problem with logistic reward model) that does not depend polynomially on $S$ (the known upper bound on the size of the model parameters, i.e. $\|\theta^\star\| \leq S$, See Section 2) [3]. RS-GLinCB is, to our knowledge, the first CB algorithm for generalized linear reward models that is both computationally efficient (amortized $O(\log T)$ computation per round) and incurs optimal regret. We also perform experiments in Section 5 that validate its superiority both in terms of regret and computational efficiency in comparison to other baseline algorithms proposed in [14] and [6].

## 1.2 Important Remarks on Contributions and Comparison with Prior Work

*Remark* 1.1 ($\kappa$-**independence**). For both B-GLinCB and RS-GLinCB, our regret guarantees are free of $\kappa$ (in their leading term), an instance-dependent parameter that can be exponential in the size of the unknown parameter vector, i.e., $\|\theta^\star\|$ (See Section 2 for definition). Our contribution in this regard is two-fold. Not only do we prove $\kappa$-independent regret guarantees under the limited adaptivity constraint, we also characterize a broad class of generalized linear reward models for which a $\kappa$-independent regret guarantee can be achieved. Specifically, our results imply that the CB problem with generalized linear reward models originally proposed in [8] and subsequently studied in literature [19, 14, 24] admits a $\kappa$-independent regret.

---

[3]This requires a non-convex projection. We discuss its convex relaxation in Appendix E.

*Remark* 1.2 (**Computational efficiency**). Efforts to reduce the total time complexity to be linear in $T$ have been active in the CB literature with generealized linear rewards models. For e.g., [14] recently devised computationally efficient algorithms but they suffer from regret dependence on $\kappa$. Optimal ($\kappa$-independent) guarantees were recently achieved for logistic reward models [6, 2], and the algorithms were subsequently made computationally efficient in [7, 28]. However, the techniques involved rely heavily on the structure of the logistic model and do not easily extend to more general models. To the best of our knowledge, ours is the first work that achieves optimal $\kappa$-independent regret guarantees for bounded generalized linear reward models while remaining computationally efficient[4].

*Remark* 1.3 (**Self Concordance of bounded GLMs**). In order to prove $\kappa$-independent regret guarantees, we prove a key result about self concordance of bounded (almost surely) generalized linear models (Definition 2.1) in Lemma 2.2. This result was postulated in [8] for GLMs (with the same definition as ours), but no proof was provided. While [6, 7] partially tackled this issue for logistic reward models[5], in our work, we prove self concordance for much more general generalized linear models.

## 2 Notations and Preliminaries

**Notations:** A policy $\pi$ is a function that maps any given arm set $\mathcal{X}$ to a probability distribution over the same set, i.e., $\pi(\mathcal{X}) \in \Delta(\mathcal{X})$, where $\Delta(\mathcal{X})$ is the probability simplex supported on $\mathcal{X}$. We will denote matrices in bold upper case (e.g. $\mathbf{M}$). $\|x\|$ denotes the $\ell_2$ norm of vector $x$. We write $\|x\|_{\mathbf{M}}$ to denote $\sqrt{x^\top \mathbf{M} x}$ for a positive semi-definite matrix $\mathbf{M}$ and vector $x$. For any two real numbers $a$ and $b$, we denote by $a \wedge b$ the minimum of $a$ and $b$. Throughout, $\widetilde{O}(\cdot)$ denotes big-O notation but suppresses log factors in all relevant parameters. For $m, n \in \mathbb{N}$ with $m < n$, we denote the set $\{1, \ldots, n\}$ by $[n]$ and $\{m, \ldots, n\}$ by $[m, n]$.

**Definition 2.1** (GLM). A Generalized Linear Model or GLM with parameter vector $\theta^\star \in \mathbb{R}^d$ is a real valued random variable $r$ that belongs to the exponential family with density function

$$\mathbb{P}(r \mid x) = \exp\left(r \cdot \langle x, \theta^* \rangle - b\left(\langle x, \theta^* \rangle\right) + c\left(r\right)\right)$$

Function $b$ (called the log-partition function) is assumed to be twice differentiable and $\dot{b}$ is assumed to be monotone. Further, we assume that $r \in [0, R]$ almost surely for some known $R \in \mathbb{R}$.

Important properties of GLMs such as $\mathbb{E}[r \mid x] = \dot{b}(\langle x, \theta^\star \rangle)$ and variance $\mathbb{V}[r \mid x] = \ddot{b}(\langle x, \theta^\star \rangle)$ are detailed in Appendix C. We define the link function $\mu$ as $\mu\left(\langle x, \theta^* \rangle\right) := \mathbb{E}[r \mid x]$. Thus, $\mu$ is also monotone. We now present a key Lemma on GLMs (see Appendix C for details) that enables us to achieve optimal regret guarantees for our algorithms designed in Sections 3 and 4.

**Lemma 2.2** (Self-Concordance of GLMs). *For any GLM supported on $[0, R]$ almost surely, the link function $\mu(\cdot)$ satisfies $|\ddot{\mu}(z)| \leq R\dot{\mu}(z)$, for all $z \in \mathbb{R}$.*

Next we describe the two CB problems with GLM rewards that we address in this paper. Let $T \in \mathbb{N}$ be the total number of rounds. At round $t \in [T]$, we receive an arm set $\mathcal{X}_t \subset \mathbb{R}^d$, with number of arms $K = |\mathcal{X}_t|$ and must select an arm $x_t \in \mathcal{X}_t$. Following this, we receive a reward $r_t$ sampled from the GLM distribution $\mathbb{P}(r|x_t)$ with unknown $\theta^*$.

**Problem 1:** In this problem we assume that at each round $t$, the set of arms $\mathcal{X}_t \subset \mathbb{R}^d$ is drawn from an unknown distribution $\mathcal{D}$. Further, we assume the constraints of limited adaptivity setting **M1**, i.e., the algorithm is given a budget $M \in \mathbb{N}$ and needs to decide upfront the $M$ rounds at which it will update its policy. Let $\text{supp}(\mathcal{D})$ denote the support of distribution $\mathcal{D}$. We want to design an algorithm that minimizes the expected cumulative regret given as

$$\mathbf{R}_T = \mathbb{E}\left[\sum_{t=1}^{T} \max_{x \in \mathcal{X}_t} \mu\left(\langle x, \theta^* \rangle\right) - \sum_{t=1}^{T} \mu\left(\langle x_t, \theta^* \rangle\right)\right]$$

---

[4]While RS-GLinCB and B-GLinCB have total running time of $\widetilde{O}(T)$, their per-round complexity can reach $O(T)$. This stands in contrast to [7], which maintains efficiency in both total and per-round time complexity.

[5]In [5], a claim about $\kappa$ independent regret for all generalized linear models with bounded $\theta^*$ was made; however, we can construct counterexamples to this claim (see Appendix C, Remark C.5).

Here, the expectation is taken over the randomness of the algorithm, the distribution of rewards $r_t$, and the distribution of the arm set $\mathcal{D}$.

**Problem 2:** In this problem we do not make any assumptions on the arm feature vectors, i.e., the arm vectors can be adversarially chosen. However, we assume the constraints of limited adaptivity setting **M2**, i.e., the algorithm is given a budget $M \in \mathbb{N}$ and needs to adaptively decide the $M$ rounds at which it will update its policy (during its course). We want to design an algorithm that minimizes the cumulative regret given as

$$\mathbf{R}_T = \sum_{t=1}^{T} \max_{x \in \mathcal{X}_t} \mu\left(\langle x, \theta^* \rangle\right) - \sum_{t=1}^{T} \mu\left(\langle x_t, \theta^* \rangle\right)$$

Finally, for both the problems, the Maximum Likelihood Estimator (MLE) of $\theta^*$ can be calculated by minimizing the sum of the log-losses. The log-loss is defined for any given arm $x$, its (stochastic) reward $r$ and vector $\theta \in \mathbb{R}^d$ (as the estimator of the true unknown $\theta^*$) as follows: $\ell(\theta, x, r) := -r \cdot \langle x, \theta \rangle + \int_0^{\langle x, \theta \rangle} \mu(z)dz$. After $t$ rounds, the MLE $\widehat{\theta}$ is computed as $\widehat{\theta} = \arg\min_\theta \sum_{s=1}^{t} \ell(\theta, x_s, r_s)$.

## 2.1 Instance Dependent Non-Linearity Parameters

As in prior works [6, 7], we define instance dependent parameters that capture non-linearity of the underlying instance and critically impact our algorithm design. The performance of Algorithm 1 (B-GLinCB) that solves Problem 1, can be quantified using three such parameters that are defined using the derivative of the link function $\dot{\mu}(\cdot)$. Specifically, for any arm set $\mathcal{X}$, write optimal arm $x^* = \arg\max_{x \in \mathcal{X}} \mu\left(\langle x, \theta^* \rangle\right)$ and define,

$$\kappa := \max_{\mathcal{X} \in \mathtt{supp}(\mathcal{D})} \max_{x \in \mathcal{X}} \frac{1}{\dot{\mu}\left(\langle x, \theta^* \rangle\right)}, \quad \frac{1}{\kappa^*} := \max_{\mathcal{X} \in \mathtt{supp}(\mathcal{D})} \dot{\mu}\left(\langle x^*, \theta^* \rangle\right), \quad \frac{1}{\widehat{\kappa}} := \mathbb{E}_{\mathcal{X} \sim \mathcal{D}}\left[\dot{\mu}\left(\langle x^*, \theta^* \rangle\right)\right] \quad (1)$$

*Remark* 2.3. These quantities feature prominently in our regret analysis of Algorithm 1. In particular, the dominant term in our regret bound scales as $O(\sqrt{T/\kappa^*})$. We also note that $\widehat{\kappa} \geq \kappa^*$; in fact, for specific distributions $\mathcal{D}$, the gap between them can be significant. Hence, we also provide a regret upper bound of $O(\sqrt{T/\widehat{\kappa}})$. In this latter case, however, we incur a worse dependence on $d$. Section 3 provides a quantified form of this trade-off.

Algorithm 2 (RS-GLinCB) that solves Problem 2, requires another such non-linearity parameter $\kappa$[6], defined as,

$$\kappa := \max_{x \in \cup_{t=1}^T \mathcal{X}_t} \frac{1}{\dot{\mu}\left(\langle x, \theta^* \rangle\right)} \quad (2)$$

We note that, here, $\kappa$ is defined considering the parameter vector $\theta^*$ in contrast to prior work on logistic bandits [7], where its definition involved a maximization over all vectors $\theta$ with $\|\theta\| \leq S$ (known upper bound of $\|\theta^\star\|$). Hence, $\kappa$ as defined here is potentially much smaller and can lead to lower regret, compared to prior works. Standard to the CB literature with GLM rewards, we will assume that tight upper bounds on these parameters is known to the algorithms.

**Assumption 2.4.** We make the following additional assumptions which are standard for the CB problem with linear or GLM reward models.

- For every round $t \in [T]$, and each arm $x \in \mathcal{X}_t$, $\|x\| \leq 1$.

- Let $\theta^*$ be the unknown parameter of the GLM reward, then $\|\theta^*\| \leq S$ for a known constant $S$.

## 2.2 Optimal Design Policies

**G-optimal Design** Given an arm set $\mathcal{X}$, the G-OPTIMAL DESIGN policy $\pi_G$ is the solution of the following optimization problem: $\arg\min_{\lambda \in \Delta(\mathcal{X})} \max_{x \in \mathcal{X}} \|x\|_{\mathbf{U}(\lambda)^{-1}}^2$, where $\mathbf{U}(\lambda) = \mathbb{E}_{x \sim \lambda}[xx^\mathsf{T}]$. Now consider the following optimization problem, also known as the D-optimal design problem: $\max_{\lambda \in \Delta(\mathcal{X})} \log \mathsf{Det}(\mathbf{U}(\lambda))$. This is a concave maximization problem as opposed to the G-optimal design which is non-convex. We have the following equivalence theorem due to Kiefer and Wolfowitz [15]:

---

[6]We overload the notation to match that in the literature. $\kappa$ in the context of Problem 1 is defined via (1), while $\kappa$ in the context of Problem 2 by (2).

---

**Algorithm 1** `B-GLinCB`: Batched Generalized Linear Bandits Algorithm

---

**Input:** Number of batches $M$ and horizon of play $T$.

1: Initialize batches $\mathcal{T}_1, \ldots, \mathcal{T}_M$, as defined in equation (3), and set $\lambda := 20Rd \log T$.
2: **for** rounds $t \in \mathcal{T}_1$ **do**
3:      Observe arm set $\mathcal{X}_t$, sample arm $x_t \sim \pi_G(\mathcal{X}_t)$, and observe reward $r_t$.
4: Compute $\widehat{\theta}_w = \arg\min_\theta \sum_{s \in \mathcal{T}_1} \ell(\theta, x_s, r_s)$ and matrix $\mathbf{V} = \lambda\mathbf{I} + \sum_{t \in \mathcal{T}_1} x_t x_t^\mathsf{T}$.
5: Initialize policy $\pi_1$ as G-OPTIMAL DESIGN.
6: **for** batches $k = 2$ to $M$ **do**
7:      **for** each round $t \in \mathcal{T}_k$ **do**
8:          Observe arm set $\mathcal{X}_t$.
9:          **for** $j = 1$ to $k - 1$ **do**
10:             Update arm set $\mathcal{X}_t \leftarrow \mathcal{X}_t \setminus \{x \in \mathcal{X}_t : UCB_j(x) < \max_{y \in \mathcal{X}_t} LCB_j(y)\}$.
11:          Scale $\mathcal{X}_t$, as in (4), to obtain $\widetilde{\mathcal{X}_t}$. , then sample $x_t \sim \pi_{k-1}\left(\widetilde{\mathcal{X}_t}\right)$.
12:      Equally divide $\mathcal{T}_k$ into two sets $\mathcal{A}$ and $\mathcal{B}$.
13:      Define $\mathbf{H}_k = \lambda\mathbf{I} + \sum_{t \in \mathcal{A}} \frac{\dot{\mu}(\langle x, \widehat{\theta}_w \rangle)}{\beta(x_t)} x_t x_t^\mathsf{T}$, and $\widehat{\theta}_k = \arg\min_\theta \sum_{s \in \mathcal{A}} \ell(\theta, x_s, r_s)$.
14:      Compute DISTRIBUTIONAL OPTIMAL DESIGN policy $\pi_k$ using the arm sets $\{\mathcal{X}_t\}_{t \in \mathcal{B}}$.

---

**Lemma 2.5** (Keifer-Wolfowitz). *Let $\mathcal{X} \subset \mathbb{R}^d$ be any set of arms and $\mathbf{W}_G$ be the expected design matrix, defined as $\mathbf{W}_G := \mathbb{E}_{x \sim \pi_G(\mathcal{X})}\left[xx^\mathsf{T}\right]$, with $\pi_G(\mathcal{X})$ as the solution to the D-optimal design problem. Then, $\pi_G(\mathcal{X})$ also solves the G-optimal design problem, and for all $x \in \mathcal{X}$, $\|x\|_{\mathbf{W}_G^{-1}}^2 \leq d$.*

**Distributional optimal design** Notably, the upper bound on $\|x\|_{\mathbf{W}_G^{-1}}$ specified in Lemma 2.5 holds only for the arms $x$ in $\mathcal{X}$. When the arm set $\mathcal{X}_t$ varies from round to round, securing a guarantee analogous to Lemma 2.5 is generally challenging. Nonetheless, when the arm sets $\mathcal{X}_t$ are drawn from a distribution, it is possible to extend the guarantee, albeit with a worse dependence on $d$; see Section A.5 in Appendix A. Improving this dependence motivates the need of studying DISTRIBUTIONAL OPTIMAL DESIGN and towards this we utilize the results of [23].

The distributional optimal design policy is defined using a collection of tuples $\mathcal{M} = \{(p_i, \mathbf{M}_i) : p_1, \ldots, p_n \geq 0$ and $\sum_i p_i = 1\}$, wherein each $\mathbf{M}_i$ is a $d \times d$ positive semi-definite matrix and $n \leq 4d \log d$. The collection $\mathcal{M}$ is detailed next. Let $\text{softmax}_\alpha(\{s_1, \ldots, s_k\})$ denote the probability distribution where the $i^{th}$ element is sampled with probability $\frac{s_i^\alpha}{\sum_{j=1}^k s_j^\alpha}$. For a specific $\mathcal{M} = \{(p_i, \mathbf{M}_i)\}_{i=1}^n$, and each $i \in [n]$ write $\pi_{\mathbf{M}_i}(\mathcal{X}) = \text{softmax}_\alpha(\{\|x\|_{\mathbf{M}_i}^2 : x \in \mathcal{X}\})$. Finally, with $\pi_G$ as the G-OPTIMAL DESIGN policy (Section 2.2), we define the DISTRIBUTIONAL OPTIMAL DESIGN policy $\pi$ as

$$\pi(\mathcal{X}) = \begin{cases} \pi_G(\mathcal{X}) & \text{with probability } 1/2 \\ \pi_{\mathbf{M}_i}(\mathcal{X}) & \text{with probability } p_i/2 \end{cases}$$

Given a collection of arm sets $\{\mathcal{X}_1, \ldots, \mathcal{X}_s\}$ (called *core set*) sampled from the distribution $\mathcal{D}$, we utilize Algorithm 2 of [23] to find the collection $\mathcal{M}$; see Algorithm 4 of [23]. Overall, the computed $\mathcal{M}$ induces a policy $\pi$ that upholds the following guarantee.

**Lemma 2.6** (Theorem 5, [23]). *Let $\pi$ be the DISTRIBUTIONAL OPTIMAL DESIGN policy that has been learnt from $s$ independent samples $\mathcal{X}_1, \ldots \mathcal{X}_s \sim \mathcal{D}$. Also, let $\mathbf{W}$ denote the expected design matrix, $\mathbf{W} = \mathbb{E}_{\mathcal{X} \sim \mathcal{D}}\left[\mathbb{E}_{x \sim \pi(\mathcal{X})}\left[xx^\mathsf{T} \mid \mathcal{X}\right]\right]$. Then,*

$$\mathbb{P}\left\{\mathbb{E}_{\mathcal{X} \sim \mathcal{D}}\left[\max_{x \in \mathcal{X}} \|x\|_{\mathbf{W}^{-1}}\right] \leq O\left(\sqrt{d \log d}\right)\right\} \geq 1 - \exp\left(O\left(d^4 \log^2 d\right) - sd^{-12} \cdot 2^{-16}\right).$$

## 3 `B-GLinCB`

In this section, we present `B-GLinCB` (Algorithm 1) that solves **Problem 1** described in Section 2, which enforces constraints of limited adaptivity setting **M1**. Given limited adaptivity budget $M \in \mathbb{N}$, our algorithm first computes the batch length for each of the $M$ batches (i.e., determine rounds where

the policy remains constant). We build upon the batch length construction in [9]; however, the first batch is chosen to be $\kappa$ dependent which crucially helps in removing $\kappa$ from the leading term in the regret. [7]

**Batch Lengths**: For each batch $k \in [M]$, let $\mathcal{T}_k$ denote all the rounds within the $k^{th}$ batch. We will refer to the first batch $\mathcal{T}_1$ as the warm-up batch. The batch lengths $\tau_k := |\mathcal{T}_k|$, $k \in [M]$ are calculated as follows:

$$\tau_1 := \left( \frac{\sqrt{\kappa}\, e^{3S} d^2 \gamma^2}{S} \alpha \right)^{2/3}, \quad \tau_2 := \alpha, \quad \tau_k := \alpha \sqrt{\tau_{k-1}}, \text{ for } k \in [3, M] \tag{3}$$

where $\gamma := 30RS\sqrt{d \log T}$ [8] and $\alpha = T^{\frac{1}{2(1-2^{-M+1})}}$ if $M \leq \log \log T$ and $\alpha = 2\sqrt{T}$ otherwise.

During the warm-up batch (Lines 2, 3), the algorithm follows the G-OPTIMAL DESIGN policy, $\pi_G$. At the end of the warm-up batch (Line 4), the algorithm computes the Maximum Likelihood Estimate (MLE), $\widehat{\theta}_w$, of $\theta^*$ [9], and design matrix $\mathbf{V} := \sum_{t \in \mathcal{T}_1} x_t x_t^\mathsf{T} + \lambda \mathbf{I}$, with parameter $\lambda = 20Rd \log T$.

Now, for each batch $k \geq 2$ and every round $t \in \mathcal{T}_k$, the algorithm updates $\mathcal{X}_t$ by eliminating arms from it using the confidence bounds (see Equation (7)) computed in the previous batches (Line 10). The algorithm next computes $\widetilde{\mathcal{X}}_t$, a scaled version of $\mathcal{X}_t$, as follows, with $\beta(x)$ define in equation (5),

$$\widetilde{\mathcal{X}}_t := \left\{ \sqrt{\dot\mu(\langle x, \widehat{\theta}_w \rangle)/\beta(x)} \; x : \; x \in \mathcal{X}_t \right\}. \tag{4}$$

Finally, we use the distributional optimal design policy $\pi_k$, on the scaled arm set $\widetilde{\mathcal{X}}_t$, to sample the next arm (Line 11). At the end of every batch, we equally divide the batch $\mathcal{T}_k$ into two sets $\mathcal{A}$ and $\mathcal{B}$. We use samples from $\mathcal{A}$ to compute the estimator $\widehat{\theta}_k$ and the scaled design matrix $\mathbf{H}_k$. The rounds in $\mathcal{B}$ are used to compute $\pi_{k+1}$, the distributional optimal design policy for the next batch. It is important to note while the policy $\pi_k$ is utilized in each round (Line 11) to draw arms, it is updated (to $\pi_{k+1}$) only at the end of the batch. Hence, conforming to setting **M1**, the algorithm updates the selection policy at $M$ rounds that were decided upfront.

**Confidence Bounds:** The scaled design matrix $\mathbf{H}_k$, an estimator of the Hessian, is computed at the end of each batch $k \in 2, \ldots, M$ (Line 13):

$$\mathbf{H}_k = \sum_{t \in \mathcal{A}} \left( \dot\mu(\langle x_t, \widehat{\theta}_w \rangle)/\beta(x_t) \right) x_t x_t^\mathsf{T} + \lambda \mathbf{I}, \quad \text{where} \quad \beta(x) = \exp\left( R \min \left\{ 2S, \gamma\sqrt{\kappa} \|x\|_{\mathbf{V}^{-1}} \right\} \right) \tag{5}$$

where $\mathcal{A}$ is the first half of $\mathcal{T}_k$. Using this, we define the upper and lower confidence bounds ($UCB_k$ and $LCB_k$) computed at the end of batch $\mathcal{T}_k$:

$$UCB_k(x) := \begin{cases} \langle x, \widehat{\theta}_w \rangle + \gamma\sqrt{\kappa} \|x\|_{\mathbf{V}^{-1}} & k = 1 \\ \langle x, \widehat{\theta}_k \rangle + \gamma \|x\|_{\mathbf{H}_k^{-1}} & k > 1 \end{cases}, \tag{6}$$

$$LCB_k(x) := \begin{cases} \langle x, \widehat{\theta}_w \rangle - \gamma\sqrt{\kappa} \|x\|_{\mathbf{V}^{-1}} & k = 1 \\ \langle x, \widehat{\theta}_k \rangle - \gamma \|x\|_{\mathbf{H}_k^{-1}} & k > 1 \end{cases} \tag{7}$$

*Remark* 3.1. The confidence bounds employed by the algorithm exhibit a significant distinction between the first batch and subsequent batches. While the first batch's bounds are influenced by the parameter $\kappa$, subsequent batches utilize $\kappa$-independent bounds. This difference arises from the use of the standard design matrix $\mathbf{V}$ in the first batch and a scaled design matrix $\mathbf{H}_k$ (equation 5) in later batches, leveraging the self-concordance property of GLM rewards to achieve $\kappa$-independence. Notably, the first batch's confidence bounds influence the scaling factor $\beta(x)$ in later batches, creating a trade-off (addressed in the regret analysis in Appendix A) where an inaccurate estimate of $\widehat{\theta}_w$ can exponentially increase the scaling factor and confidence bounds.

In Theorem 3.2 and Corollary 3.3, we present our regret guarantee for B-GLinCB. Detailed proofs for both are provided in Appendix A. The computational efficiency of B-GLinCB is discussed in Appendix D.

---

[7] We note that in case $\kappa$ is unknown, any known upper bound on $\kappa$ suffices for the algorithm.

[8] Recall that $R$ provides an upper bound on the stochastic rewards and $S$ is an upper bound on the norm of $\theta^*$.

[9] In case the MLE lies outside the set $\{\theta^* : \|\theta^*\| \leq S\}$, we apply the projection step detailed in Appendix E.

---
**Algorithm 2** `RS-GLinCB`: Rarely-Switching GLM Bandit Algorithm
---
1: Initialize: $\mathbf{V} = \mathbf{H}_1 = \lambda\mathbf{I}, \mathcal{T}_o = \emptyset, \tau = 1, \lambda \coloneqq d\log(T/\delta)/R^2$ and $\gamma \coloneqq 25RS\sqrt{d\log\left(\frac{T}{\delta}\right)}$.
2: **for** rounds $t = 1, \ldots, T$ **do**
3:     Observe arm set $\mathcal{X}_t$.
4:     **if** $\max_{x\in\mathcal{X}_t}\|x\|^2_{\mathbf{V}^{-1}} \geq 1/(\gamma^2\kappa R^2)$ **then**     // Switching Criterion I
5:         Select $x_t = \arg\max_{x\in\mathcal{X}_t}\|x\|_{\mathbf{V}^{-1}}$ and observe reward $r_t$.
6:         Update $\mathcal{T}_o \leftarrow \mathcal{T}_o \cup \{t\}, \mathbf{V} \leftarrow \mathbf{V} + x_t x_t^\mathsf{T}$ and $\mathbf{H}_{t+1} \leftarrow \mathbf{H}_t$.
7:         Compute $\widehat{\theta}_o = \arg\min_\theta \sum_{s\in\mathcal{T}_o}\ell(\theta, x_s, r_s) + \frac{\lambda}{2}\|\theta\|_2^2$.
8:     **else**
9:         **if** $\det(\mathbf{H}_t) > 2\det(\mathbf{H}_\tau)$ **then**     // Switching Criterion II
10:           Set $\tau = t$ and $\widetilde{\theta} \leftarrow \arg\min_\theta \sum_{s\in[t-1]\setminus\mathcal{T}_o}\ell(\theta, x_s, r_s) + \frac{\lambda}{2}\|\theta\|_2^2$ and
11:           $\widehat{\theta}_\tau \leftarrow \texttt{Project}(\widetilde{\theta})$.
12:         Update $\mathcal{X}_t \leftarrow \mathcal{X}_t \setminus \{x \in \mathcal{X}_t : UCB_o(x) < \max_{z\in\mathcal{X}_t} LCB_o(z)\}$.
13:         Select $x_t = \arg\max_{x\in\mathcal{X}_t} UCB(x, \mathbf{H}_\tau, \widehat{\theta}_\tau)$ and observe reward $r_t$.
14:         Update $\mathbf{H}_{t+1} \leftarrow \mathbf{H}_t + \frac{\dot{\mu}(\langle x_t, \widehat{\theta}_o\rangle)}{e}x_t x_t^\mathsf{T}$.
---

**Theorem 3.2.** *Algorithm 1 (*`B-GLinCB`*) incurs regret* $\mathrm{R}_T \leq (\mathrm{R}_1 + \mathrm{R}_2)\log\log T$ [10]*, where*

$$\mathrm{R}_1 = O\left(RSd\left(\sqrt{\frac{d}{\widehat{\kappa}}} \wedge \sqrt{\frac{1}{\kappa^*}}\right)T^{\overline{2(1-2^{1-M})}}\log T\right) \text{ and }$$

$$\mathrm{R}_2 = O\left(\kappa^{1/3}d^2 e^{2S}(RS\log T)^{2/3}T^{\overline{3(1-2^{1-M})}}\right).$$

**Corollary 3.3.** *When the number of batches* $M \geq \log\log T$*, Algorithm 1 achieves a regret bound of*

$$\mathrm{R}_T \leq \widetilde{O}\left(\left(\sqrt{\frac{d}{\widehat{\kappa}}} \wedge \sqrt{\frac{1}{\kappa^*}}\right)dRS\sqrt{T} + d^2 e^{2S}(S^2 R^2\kappa T)^{1/3}\right).$$

*Remark* 3.4. Scaling the arm set (as in (4)) for optimal design is a crucial aspect of our algorithm, allowing us to obtain tight estimates of $\dot{\mu}\left(\langle x, \theta^*\rangle\right)$ (see Lemma A.10). This result relies on multiple novel ideas and techniques, including self-concordance for GLMs, matrix concentration, Bernstein-type concentration for the canonical exponential family (Lemma A.1), and application of distributional optimal design on scaled arm set.

*Remark* 3.5. The $\kappa$-dependent batch construction is a crucial feature of our algorithm, enabling effective estimation of $\dot{\mu}(\langle x, \theta^*\rangle)$ at the end of the first batch. Since the first batch incurs regret linear in its length, achieving a $\kappa$-independent guarantee requires the first batch to be $o(\sqrt{T})$. We demonstrate that choosing $\tau_1 = O(T^{\frac{1}{3}})$ is sufficient for this purpose (see Appendix A).

## 4   `RS-GLinCB`

In this section we present `RS-GLinCB` (Algorithm 2) that solves **Problem 2** described in Section 2, which enforces constraints of limited adaptivity setting **M2**. This algorithm incorporates a novel switching criterion (Line 4), extending the determinant-doubling approach of [1]. Additionally, we introduce an arm-elimination step (Line 12) to obtain tighter regret guarantees. Throughout this section, we set $\lambda = d\log(T/\delta)/R^2$ and $\gamma = 25RS\sqrt{d\log(T/\delta)}$.

At round $t$, on receiving an arm set $\mathcal{X}_t$, `RS-GLinCB` first checks the Switching Criterion I (Line 4). This criterion checks whether for any arm $x \in \mathcal{X}_t$ the quantity $\|x\|_{\mathbf{V}^{-1}}$ is greater than a carefully chosen $\kappa$-dependent threshold. Here $\mathbf{V}$ is the design matrix corresponding to all arms that have been played in the rounds in $\mathcal{T}_o$ ($\coloneqq$ the set of rounds preceding round $t$, where Switching Criterion I was triggered). Under this criterion the arm that maximizes $\|x\|_{\mathbf{V}^{-1}}$ is played (call this arm $x_t$) and the corresponding reward is obtained. Subsequently in Line 6, the set $\mathcal{T}_o$ is updated to include $t$; the

---
[10] Note that $\mathrm{R}_T$ is expected regret. See the 2.

design matrix $\mathbf{V}$ is updated as $\mathbf{V} \leftarrow \mathbf{V} + x_t x_t^\mathsf{T}$; and the scaled design matrix $\mathbf{H}_{t+1}$ is set to $\mathbf{H}_t$. The MLE is computed (Line 7) based on the data in the rounds in $\mathcal{T}_o$ to obtain $\widehat{\theta}_o$.

When Switching Criterion I is not triggered, the algorithm first checks (Line 9) the Switching Criterion II, that is whether the determinant of the scaled design matrix $\mathbf{H}_t$ has become more than double of that of $\mathbf{H}_\tau$ (where $\tau$ is the last round before $t$ when Switching Criterion II was triggered). If Switching Criterion II is triggered at round $t$, then in Line 10, the algorithm sets $\tau \leftarrow t$ and recomputes the MLE over all the past rounds except those in $\mathcal{T}_o$ to obtain $\widetilde{\theta}$. Then $\widetilde{\theta}$ is projected into an ellipsoid around $\widehat{\theta}_o$ to obtain the estimate $\widehat{\theta}_\tau$ via the following optimization problem[11],

$$\min_\theta \left\| \sum_{s \in \mathcal{T}_o} \left( \mu\left(\langle x_s, \theta \rangle\right) - \mu(\langle x_s, \widetilde{\theta}\rangle) \right) x_s \right\|_{\mathbf{H}(\theta)} \quad \text{s.t.} \quad \left\| \theta - \widehat{\theta}_o \right\|_{\mathbf{V}} \leq \gamma\sqrt{\kappa}. \tag{8}$$

Here $\mathbf{H}(\theta) := \sum_{s \in \mathcal{T}_o} \dot{\mu}\left(\langle x_s, \theta \rangle\right) x_s x_s^\mathsf{T}$. After checking Switching Criterion II, the algorithm performs an arm elimination step (Line 12) based on the parameter estimate $\widehat{\theta}_o$ as follows: for every arm $x \in \mathcal{X}_t$, we compute $UCB_o(x) = \langle x, \widehat{\theta}_o \rangle + \gamma\sqrt{\kappa}\|x\|_{\mathbf{V}^{-1}}$ and $LCB_o(x) = \langle x, \widehat{\theta}_o \rangle - \gamma\sqrt{\kappa}\|x\|_{\mathbf{V}^{-1}}$[12]. Then, $\mathcal{X}_t$ is updated by eliminating from it the arms with $UCB_o(\cdot)$ less than the highest $LCB_o(\cdot)$. For arms in the reduced arm set $\mathcal{X}_t$, RS-GLinCB computes the index $UCB(x, \mathbf{H}_\tau, \widehat{\theta}_\tau) := \langle x, \widehat{\theta}_\tau \rangle + 150\|x\|_{\mathbf{H}_\tau^{-1}}\sqrt{d\log\left(T/\delta\right)}$, and plays the arm $x_t$ with the highest index (Line 13). After observing the subsequent reward $r_t$, the algorithm updates the scaled design matrix $\mathbf{H}_t$ (Line 14) as follows: $\mathbf{H}_{t+1} \leftarrow \mathbf{H}_t + (\dot{\mu}(\langle x_t, \widehat{\theta}_o \rangle)/e)x_t x_t^\mathsf{T}$. With this, the round $t$ ends and the algorithm moves to the next round. Next, in Lemma 4.1 and Theorem 4.2 we present the guarantees on number of policy updates and regret, respectively, for RS-GLinCB. Detailed proofs for both are provided in Appendix B.

**Lemma 4.1.** *RS-GLinCB (Algorithm 2), during its entire execution, updates its policy at most $O(R^4 S^2 \kappa d^2 \log^2(T/\delta))$ times.*

**Theorem 4.2.** *Given $\delta \in (0,1)$, with probability $\geq 1 - \delta$, the regret of RS-GLinCB (Algorithm 2) satisfies $\mathrm{R}_T = O\big(d\sqrt{\sum_{t \in [T]} \dot{\mu}\left(\langle x_t^*, \theta^* \rangle\right)}\log\left(RT/\delta\right) + \kappa d^2 R^5 S^2 \log^2\left(T/\delta\right)\big)$.*

*Remark* 4.3. Switching Criterion I is essential in delivering tight regret guarantees in the non-linear setting. Unlike existing literature [7], which relies on warm-up rounds based on observed rewards (hence heavily dependent on reward models), RS-GLinCB presents a context-dependent criterion that implicitly checks whether the estimate $\dot{\mu}(\langle x, \widehat{\theta}_o \rangle)$ is within a constant factor of $\dot{\mu}\left(\langle x, \theta^* \rangle\right)$ (see Lemmas B.3 and B.4). We show that the number of times Switching Criterion I is triggered is only $O(\kappa d^2 \log^2(T))$ (see Lemma B.11), hence incurring a small regret in these rounds.

*Remark* 4.4. Unlike [1], our determinant-doubling Switching Criterion II uses the scaled design matrix $\mathbf{H}_t$ instead of the unscaled version (similar to $\mathbf{V}$). The matrix $\mathbf{H}_t$, estimating the Hessian of the log-loss, is crucial for achieving optimal regret. This modification is crucial in extending algorithms satisfying limited adaptivity setting **M2** for the CB problem with a linear reward model to more general GLM reward models.

*Remark* 4.5. The feasible set for the optimization stated in 8 is an ellipsoid around $\widehat{\theta}_o$, which contains $\theta^*$ with high probability. Deviating from existing literature on GLM Bandits which projects the estimate into the ball set of radius $S$ ($\{\theta : \|\theta\| \leq S\}$), our projection step leads to tighter regret guarantees; notably, the leading $\sqrt{T}$ term is free of parameters $S$ (and $R$). This resolves the conjecture made in [17] regarding the possibility of obtaining $S$-free regret in the $\sqrt{T}$ term in logistic bandits.

*Remark* 4.6. The regret guarantees of the logistic bandit algorithms in [2, 17] have a second-order term that is minimum of an arm-geometry dependent quantity (see Theorem 3 of [17]) and a $\kappa$-dependent term similar to our regret guarantee. Although our analysis is not able to accommodate this arm-geometry dependent quantity, we underscore that our algorithm is computationally efficient while the above works are not. In fact, to the best of our knowledge, the other known efficient algorithms for logistic bandits [7, 28] also do not achieve the arm-geometry dependent regret term. It can be interesting to design an efficient algorithm that is able to achieve the same guarantees in the second-order regret term as in [2, 17].

---

[11]This optimization problem is non-convex. However, a convex relation of this optimization problem is detailed in Appendix E, which leads to slightly worse regret guarantees in $\mathrm{poly}(R, S)$.

[12]We note that in case $\kappa$ is unknown, any known upper bound on $\kappa$ suffices for the algorithm.

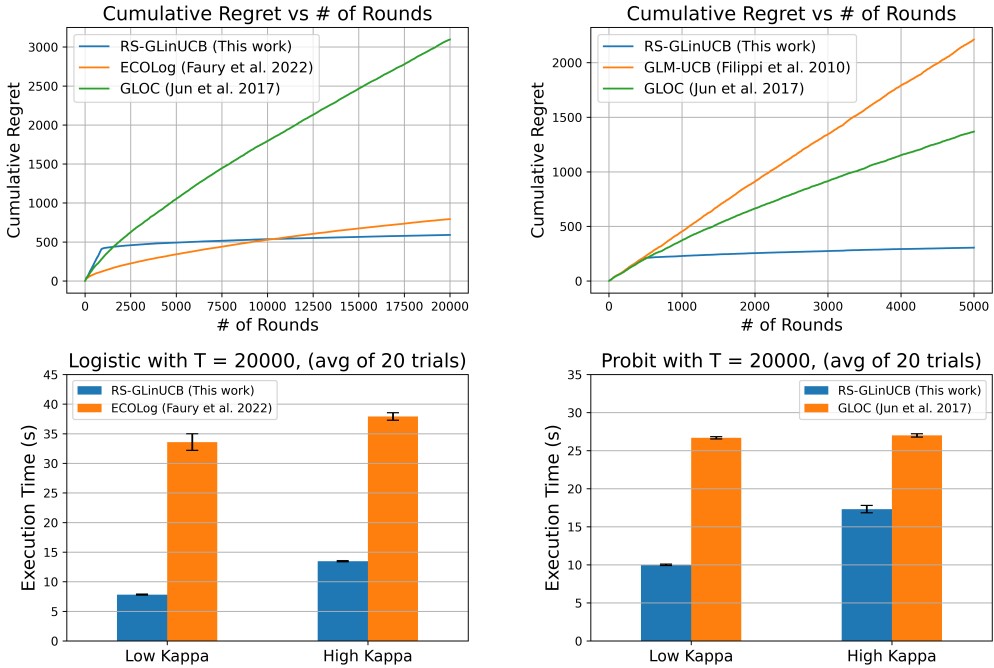

Figure 1: Top: Cumulative Regret vs. number of rounds for Logistic (left) and Probit (right) reward models. Bottom: (left) Execution times of `ECOLog` and `RS-GLinCB` for different values of $\kappa$ (low $\kappa = 9.3$ and high $\kappa = 141.6$) for Logistic rewards. (right) Execution times of `GLOC` and `RS-GLinCB` for different values of $\kappa$ (low $\kappa = 17.6$ and high $\kappa = 202.3$) for Probit rewards.

## 5 Experiments

We tested the practicality of our algorithm `RS-GLinCB` against various baselines for logistic and generalized linear bandits. For these experiments, we adjusted the Switching Criterion I threshold constant in `RS-GLinCB` to $0.01$ and used data from both Switching Criteria (I and II) rounds to estimate $\widetilde{\theta}$. These modifications do not affect the overall efficiency as $\widetilde{\theta}$ is calculated only $O(\log(T))$ times. The experiment code is available at `https://github.com/nirjhar-das/GLBandit_Limited_Adaptivity`.

**Logistic.** We compared `RS-GLinCB` against `ECOLog` [7] and `GLOC` [14], the only algorithms with overall time complexity $\tilde{O}(T)$ for this setting. The dimension was set to $d = 5$, number of arms per round to $K = 20$, and $\theta^*$ was sampled from a $d$-dimensional sphere of radius $S = 5$. Arms were sampled uniformly from the $d$-dimensional unit ball. We ran simulations for $T = 20,000$ rounds, repeating them 10 times. `RS-GLinCB` showed the smallest regret with a flattened regret curve, as seen in Fig. 1 (top-left).

**Probit.** For the probit reward model, we compared `RS-GLinCB` against `GLOC` and `GLM-UCB` [8]. The dimension was set to $d = 5$ and number of arms per round to $K = 20$. $\theta^*$ was sampled from a $d$-dimensional sphere of radius $S = 3$. Arm features were generated similarly as in the logistic bandit simulation. We ran simulations for $T = 5,000$ rounds, repeating them 10 times. `RS-GLinCB` outperformed both baselines, as shown in Fig. 1 (top-right).

**Comparing Execution Times.** We compared the execution times of `RS-GLinCB` and `ECOLog`. We created two logistic bandit instances with $d = 5$ and $K = 20$, and different $\kappa$ values. We ran both algorithms for $T = 20,000$ rounds, repeating each run 20 times. For low $\kappa$, `RS-GLinCB` took about one-fifth of the time of `ECOLog`, and for high $\kappa$, slightly more than one-third, as seen in Fig. 1 (left-bottom). This demonstrates that `RS-GLinCB` has a significantly lower computational overhead compared to `ECOLog`. We also compared the execution times of `RS-GLinCB` and `GLOC` under the probit reward model, creating two bandit instances with $d = 5$ and $K = 20$, but with differing $\kappa$. We ran both algorithms for $T = 20,000$ rounds, repeating each run 20 times. The result is shown in Fig. 1 (bottom-right). We observe that for low $\kappa$, `RS-GLinCB` takes less than half time of `GLOC` while for high $\kappa$, it takes about two-third time of `GLOC`. A more detailed discussion of these experiments is provided in Appendix D.

# 6 Conclusion and Future Work

The Contextual Bandit problem with GLM rewards is a ubiquitous framework for studying online decision-making with non-linear rewards. We study this problem with a focus on limited adaptivity. In particular, we design algorithms `B-GLinCB` and `RS-GLinCB` that obtain optimal regret guarantees for two prevalent limited adaptivity settings **M1** and **M2** respectively. A key feature of our guarantees are that their leading terms are independent of an instance dependent parameter $\kappa$ that captures non-linearity. To the best of our knowledge, our paper provides the first algorithms for the `CB` problem with GLM rewards under limited adaptivity (and otherwise) that achieve $\kappa$-independent regret. The regret guarantee of `RS-GLinCB`, not only aligns with the best-known guarantees for Logistic Bandits but enhances them by removing the dependence on $S$ (upper bound on $\|\theta^*\|$) in the leading term of the regret and therefore resolves a conjecture in [17]. The batch learning algorithm `B-GLinCB`, for $M = \Omega(\log(\log T))$, achieves a regret of $\widetilde{O}\left(dRS\left(\sqrt{d/\widehat{\kappa}} \wedge \sqrt{1/\kappa^*}\right)\sqrt{T}\right)$. We believe that the dependence on $d$ along with the $\widehat{\kappa}$ term is not tight and improving the dependence is a relevant direction for future work.

## Acknowledgments and Disclosure of Funding

Siddharth Barman gratefully acknowledges the support of the Walmart Center for Tech Excellence (CSR WMGT-23-0001) and a SERB Core research grant (CRG/2021/006165).

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

# Generalized Linear Bandits with Limited Adaptivity Appendix

## Table of Contents

# A Regret Analysis of `B-GLinCB`

## A.1 Additional Notation

We write **c** to denote absolute constant(s) that appears throughout our analysis. Our analysis also utilizes the following function

$$\gamma(\lambda) = 24RS \left( \sqrt{\log(T) + d} + \frac{R(\log(T) + d)}{\sqrt{\lambda}} \right) + 2S\sqrt{\lambda}. \tag{9}$$

Note that $\gamma(\lambda)$ is a 'parameterized' version of $\gamma$ (which was defined in section 3). In our proof, we present the arguments using this parameterized version. A direct minimization of the above expression in terms of $\lambda$ would not suffice since we need $\lambda$ to be sufficiently large for certain matrix concentration lemmas to hold (see Section A.5). However, later we show that setting $\lambda$ equal to $\mathbf{c}Rd\log T$ leads to the desired bounds.

We use $\widetilde{x}$ to denote the scaled versions of the arms (see Line 11 of the algorithm); in particular,

$$\widetilde{x} := \sqrt{\frac{\dot{\mu}\left(\langle x, \widehat{\theta}_w \rangle\right)}{\beta(x)}} x \tag{10}$$

Furthermore, to capture the non-linearity of the problem, we introduce the term $\phi(\lambda)$:

$$\phi(\lambda) := \frac{\sqrt{\kappa}\, e^{3S}\, \gamma(\lambda)^2}{S}.$$

Recall that the scaled data matrix $\mathbf{H}_k$ (for each batch $k$) was computed using $\widehat{\theta}_w$ as follows

$$\mathbf{H}_k = \sum_{t \in \mathcal{T}_k} \frac{\dot{\mu}\left(\langle x_j, \widehat{\theta}_w \rangle\right)}{\beta(x_t)} x_t x_t^\mathsf{T} + \lambda \mathbf{I}.$$

Following the definition of $\mathbf{H}_k$ and using the true vector $\theta^*$ we define

$$\mathbf{H}_k^* = \sum_{t \in \mathcal{T}_k} \dot{\mu}\left(\langle x_t, \theta^* \rangle\right) x_t x_t^\mathsf{T} + \lambda \mathbf{I}.$$

We will show that $\mathbf{H}_k$ dominates $\mathbf{H}_k^*$ with high probability. Furthermore, we assume that the MLE estimator $\theta^*$ obtained by minimizing the log-loss objective always satisfies $\left\|\widehat{\theta}\right\| \leq S$. In case, that's not true, one can use the non-convex projection described in Appendix E. The projected vector satisfies the same guarantees as described in the subsequent lemmas up to a multiplicative factor of 2. Hence, the assumption $\left\|\widehat{\theta}\right\| \leq S$ is non-limiting.

## A.2 Concentration Inequalities and Confidence Intervals

**Lemma A.1** (Bernstein's Inequality). *Let $X_1, \ldots, X_n$ be a sequence of independent random variables with $|X_t - \mathbb{E}[X_t]| \leq b$. Also, let sum $S := \sum_{t=1}^n (X_t - \mathbb{E}[X_t])$ and $v := \sum_{t=1}^m Var[X_t]$. Then, for any $\delta \in [0, 1]$, we have*

$$\mathbb{P}\left\{ S \geq \sqrt{2v \log \frac{1}{\delta}} + \frac{2b}{3} \log \frac{1}{\delta} \right\} \leq \delta.$$

**Lemma A.2.** *Let $\mathcal{X} = \{x_1, x_2, \ldots, x_s\} \in \mathbb{R}^d$ be a set of vectors with $\|x_t\| \leq 1$, for all $t \in [s]$, and let scalar $\lambda \geq 0$. Also, let $r_1, r_2, \ldots, r_s \in [0, R]$ be independent random variables distributed by the canonical exponential family; in particular, $\mathbb{E}[r_s] = \mu(\langle x_s, \theta^* \rangle)$ for $\theta^* \in \mathbb{R}^d$. Further, let $\widehat{\theta} = \arg\min_\theta \sum_{s=1}^t \ell(\theta, x_s, r_s)$ be the maximum likelihood estimator of $\theta^*$ and let matrix*

$$\mathbf{H}^* = \sum_{j=1}^s \dot{\mu}\left(\langle x_j, \theta^* \rangle\right) x_j x_j^\mathsf{T} + \lambda \mathbf{I}.$$

*Then, with probability at least than $1 - \frac{1}{T^2}$, the following inequality holds*

$$\left\| \theta^* - \widehat{\theta} \right\|_{\mathbf{H}^*} \le 24 RS \left( \sqrt{\log(T) + d} + \frac{R \left( \log(T) + d \right)}{\sqrt{\lambda}} \right) + 2S\sqrt{\lambda} \tag{11}$$

*Proof.* We first define the following quantities

$$\alpha(x, \theta^*, \widehat{\theta}) := \int_{v=1}^{1} \dot{\mu} \left( \langle x, \theta^* \rangle + v \langle x, \left( \widehat{\theta} - \theta^* \right) \rangle \right) dv$$

$$\mathbf{G} := \sum_{j=1}^{s} \alpha(x, \theta^*, \widehat{\theta}) x_j x_j^\mathsf{T} + \frac{\lambda}{1 + 2RS} \mathbf{I}$$

Using Lemma C.2 we have

$$\mathbf{G} \succeq \frac{1}{1 + 2RS} \mathbf{H}^* \tag{12}$$

Hence we write

$$
\begin{aligned}
\left\| \theta^* - \widehat{\theta} \right\|_{\mathbf{H}^*} &\le \sqrt{(1 + 2RS)} \left\| \theta^* - \widehat{\theta} \right\|_{\mathbf{G}} \\
&= \sqrt{1 + 2RS} \left\| \mathbf{G} \left( \theta^* - \widehat{\theta} \right) \right\|_{\mathbf{G}^{-1}} \\
&= \sqrt{1 + 2RS} \left\| \sum_{j=1}^{s} \left( \langle \theta^*, x_j \rangle - \langle \widehat{\theta}, x_j \rangle \right) \alpha(x, \theta^*, \widehat{\theta}) x_j + \frac{\lambda}{1 + 2RS} \left( \theta^* - \widehat{\theta} \right) \right\|_{\mathbf{G}^{-1}} \\
&\le \sqrt{1 + 2RS} \left\| \sum_{j=1}^{s} \left( \mu \left( \langle \theta^*, x_j \rangle \right) - \mu \left( \langle \widehat{\theta}, x_j \rangle \right) \right) x_j \right\|_{\mathbf{G}^{-1}} + \frac{\lambda}{\sqrt{1 + 2RS}} \left\| \theta^* - \widehat{\theta} \right\|_{\mathbf{G}^{-1}} \\
&\le \sqrt{1 + 2RS} \left\| \sum_{j=1}^{s} \left( \mu \left( \langle \theta^*, x_j \rangle \right) - \mu \left( \langle \widehat{\theta}, x_j \rangle \right) \right) x_j \right\|_{\mathbf{G}^{-1}} + 2S\sqrt{\lambda} \\
&\qquad\qquad\qquad \text{(since } \mathbf{G} \succeq \frac{\lambda}{1 + 2RS} \mathbf{I} \text{ and } \left\| \theta^* - \widehat{\theta} \right\|_2 \le 2S) \\
&\le 3RS \left\| \sum_{j=1}^{s} \left( \mu \left( \langle \theta^*, x_j \rangle \right) - \mu \left( \langle \widehat{\theta}, x_j \rangle \right) \right) x_j \right\|_{\mathbf{H}^{*-1}} + 2S\sqrt{\lambda} \\
&\qquad\qquad\qquad \text{(Using (12) and assuming } RS \ge 1)
\end{aligned}
$$

Now by the optimality condition on $\widehat{\theta}$ we have $\sum_{j=1}^{s} \mu \left( \langle x_j, \widehat{\theta} \rangle \right) x_j = \sum_{j=1}^{s} r_j x_j$ (see equation (3) [8]). Hence, we write

$$\left\| \sum_{j=1}^{s} \left( \mu \left( \langle \theta^*, x_j \rangle \right) - \mu \left( \langle \widehat{\theta}, x_j \rangle \right) \right) x_j \right\|_{\mathbf{H}^{*-1}} = \left\| \sum_{j=1}^{s} \left( \mu \left( \langle \theta^*, x_j \rangle \right) - r_j \right) x_j \right\|_{\mathbf{H}^{*-1}} \tag{13}$$

Let $\mathcal{B}$ denote the unit ball in $\mathbb{R}^d$. We can write

$$\left\| \sum_{j=1}^{s} \left( \mu \left( \langle \theta^*, x_j \rangle \right) - r_j \right) x_j \right\|_{\mathbf{H}^{*-1}} = \max_{y \in \mathcal{B}} \langle y, \mathbf{H}^{*-1/2} \sum_{j=1}^{s} \left( \mu \left( \langle \theta^*, x_j \rangle \right) - r_j \right) x_j \rangle$$

We construct an $\varepsilon$-net for the unit ball, denoted as $\mathcal{C}_\varepsilon$. For any $y \in \mathcal{B}$, we define $y_\varepsilon := \arg\min_{b \in \mathcal{C}_\varepsilon} \|b - y\|_2$. We can now write

$$\left\|\sum_{j=1}^s \left(\mu\left(\langle\theta^*, x_j\rangle\right) - r_j\right) x_j\right\|_{\mathbf{H}^{*-1}}$$

$$= \max_{y \in \mathcal{B}}\langle y - y_\varepsilon, \mathbf{H}^{*-1/2} \sum_{j=1}^s \left(\mu\left(\langle\theta^*, x_j\rangle\right) - r_j\right) x_j\rangle + \langle y_\varepsilon, \mathbf{H}^{*-1/2} \sum_{j=1}^s \left(\mu\left(\langle\theta^*, x_j\rangle\right) - r_j\right) x_j\rangle$$

$$\leq \max_{y \in \mathcal{B}} \|y - y_\varepsilon\|_2 \left\|\sum_{j=1}^s \left(\mu\left(\langle\theta^*, x_j\rangle\right) - r_j\right) x_j\right\|_{\mathbf{H}^{*-1}} + \langle y_\varepsilon, \mathbf{H}^{*-1/2} \sum_{j=1}^s \left(\mu\left(\langle\theta^*, x_j\rangle\right) - r_j\right) x_j\rangle$$

$$\leq \max_{y \in \mathcal{B}} \varepsilon \left\|\sum_{j=1}^s \left(\mu\left(\langle\theta^*, x_j\rangle\right) - r_j\right) x_j\right\|_{\mathbf{H}^{*-1}} + \langle y_\varepsilon, \mathbf{H}^{*-1/2} \sum_{j=1}^s \left(\mu\left(\langle\theta^*, x_j\rangle\right) - r_j\right) x_j\rangle$$

Rearranging, we obtain

$$\left\|\sum_{j=1}^s \left(\mu\left(\langle\theta^*, x_j\rangle\right) - r_j\right) x_j\right\|_{\mathbf{H}^{*-1}} \leq \frac{1}{1-\varepsilon}\langle y_\varepsilon, \mathbf{H}^{*-1/2} \sum_{j=1}^s \left(\mu\left(\langle\theta^*, x_j\rangle\right) - r_j\right) x_j\rangle$$

Next, we use Lemma A.3 (stated below) with $\delta = T^2|\mathcal{C}_\varepsilon|$ and union bound over all vectors in $\mathcal{C}_\varepsilon$. We also observe that $|\mathcal{C}_\varepsilon| \leq \left(\frac{2}{\varepsilon}\right)^d$. Substituting $\epsilon = 1/2$ and using Lemma A.3, we obtain that the following holds with probability greater than $1 - \frac{1}{T^2}$,

$$\left\|\sum_{j=1}^s \left(\mu\left(\langle\theta^*, x_j\rangle\right) - r_j\right) x_j\right\|_{\mathbf{H}^{*-1}} \leq 3\sqrt{\log\left(T^2|\mathcal{C}_\varepsilon|\right)} + \frac{4R}{3\sqrt{\lambda}}\log\left(T^2|\mathcal{C}_\varepsilon|\right)$$

$$\leq 8\left(\sqrt{\log\left(T\right) + d} + \frac{R\left(\log\left(T\right) + d\right)}{\sqrt{\lambda}}\right)$$

Substituting in equations (13), we get the desired inequality in the lemma statement. $\qquad\square$

**Lemma A.3.** *Let $y$ be a fixed vector with $\|y\| \leq 1$. Then, with the notation stated in Lemma A.2, the following inequality holds with probability at least $1 - \delta$*

$$\sum_{j=1}^s \left(\mu\left(\langle\theta^*, x_j\rangle\right) - r_j\right) y^\mathsf{T}\mathbf{H}^{*-1/2}x_j \leq \sqrt{2\log\frac{1}{\delta}} + \frac{2R}{3\sqrt{\lambda}}\log\frac{1}{\delta}.$$

*Proof.* Let us denote the $j^{th}$ term of the sum as $Z_j$. Note that each random variable $Z_j$ has variance $\text{Var}(Z_j) = \dot\mu\left(\langle x_j, \theta^*\rangle\right)\left(y^\mathsf{T}\mathbf{H}^{*-1/2}x_j\right)^2$. Hence, we have

$$\sum_{j=1}^s \text{Var}(Z_j) = \sum_{j=1}^s \dot\mu\left(\langle\theta^*, x_j\rangle\right)\left(y^\mathsf{T}\mathbf{H}^{*-1/2}x_j\right)^2$$

$$= y^\mathsf{T}y \leq 1.$$

Moreover, each $Z_j$ is at most $\frac{R}{\sqrt{\lambda}}$ (since $\|x_j\| \leq 1$, $\mathbf{H}^* \succeq \lambda\mathbf{I}$ and $r \in [0, R]$). Now applying Lemma A.1, we have

$$\mathbb{P}\left\{\sum_{j=1}^s Z_j \geq \sqrt{2\log\frac{1}{\delta}} + \frac{2R}{3\sqrt{\lambda}}\log\frac{1}{\delta}\right\} \leq \delta.$$

$\qquad\square$

**Corollary A.4.** *Let $x_1, x_2, \ldots, x_\tau$ be the sequence of arms pulled during the warm-up batch and let $\widehat{\theta}_w$ be the estimator of $\theta^*$ computed at the end of the batch. Then, for any vector $x$ and $\lambda \geq 0$ the following bound holds with probability greater than $1 - \frac{1}{T^2}$*

$$|\langle x, \theta^* - \widehat{\theta}_w \rangle| \leq \sqrt{\kappa}\, \|x\|_{\mathbf{V}^{-1}}\, \gamma(\lambda).$$

*Proof.* This result is derived directly from Lemma A.2 and the definition of $\gamma(\lambda)$ (see 9). By applying the lemma, we obtain

$$|\langle x, \theta^* - \widehat{\theta}_w \rangle| \leq \|x\|_{\mathbf{H}^{*-1}} \left\| \theta^* - \widehat{\theta}_w \right\|_{\mathbf{H}^*} \leq \|x\|_{\mathbf{H}^{*-1}}\, \gamma(\lambda)$$

Considering the definition of $\kappa$, we have $\dot{\mu}\left(\langle \mathbf{x}, \theta^* \rangle\right) \geq \frac{1}{\kappa}$. This implies that $\mathbf{H}^* \succeq \frac{1}{\kappa}\mathbf{V}$ [13] which in turn leads to the inequality $\|x\|_{\mathbf{H}^{*-1}} \leq \sqrt{\kappa}\, \|x\|_{\mathbf{V}^{-1}}$. $\qquad\square$

## A.3 Preliminary Lemmas

**Lemma A.5.** *For each batch $k \geq 2$ and the scaled data matrix $\mathbf{H}_k$ computed at the end of batch, the following bound holds with probability at least $1 - \frac{1}{T^2}$:*

$$\mathbf{H}_k \preceq \mathbf{H}_k^*.$$

*Proof.* If the event stated in Lemma A.4 holds,

From Lemma C.2, we apply the multiplicative bound on $\dot{\mu}$ to obtain

$$\dot{\mu}\left(\langle x, \widehat{\theta}_w \rangle\right) \leq \dot{\mu}\left(\langle x, \theta^* \rangle\right) \exp\left(R|\langle x, \widehat{\theta}_w - \theta^* \rangle|\right)$$

Via Corollary A.4 we have $|\langle x, \widehat{\theta}_w \rangle - \langle x, \theta^* \rangle| \leq \sqrt{\kappa}\, \|x\|_{\mathbf{V}^{-1}}\, \gamma(\lambda)$. Additionally, given that $\left\| \widehat{\theta} \right\|, \|\theta^*\| \leq S$ and $\|x\| \leq 1$ we also have $|\langle x, \widehat{\theta}_w \rangle - \langle x, \theta^* \rangle| \leq 2S$. Hence, we write

$$\dot{\mu}\left(\langle x, \widehat{\theta}_w \rangle\right) \leq \dot{\mu}\left(\langle x, \theta^* \rangle\right) \exp\left(R\min\{\sqrt{\kappa}\, \|x\|_{\mathbf{V}^{-1}}\, \gamma(\lambda), 2S\}\right)$$
$$\leq \dot{\mu}\left(\langle x, \theta^* \rangle\right) \beta(x)$$

Substituting these results into the definitions of $\mathbf{H}_k$ and $\mathbf{H}_k^*$ proves the lemma statement. $\qquad\square$

**Claim A.6.** *The Algorithm 1 runs for at most $\log\log T$ batches.*

*Proof.* When $M \leq \log\log T$ then the claim trivially holds. When $M \geq \log\log T + 1$, we define the length of the second batch, $\tau_2$, as $2\sqrt{T}$. The length of the $M^{th}$ batch is

$$\tau_M = (2\sqrt{T})^{\sum_{k=1}^{M-1} \frac{1}{2^{k-1}}}$$
$$\geq 2T\, T^{\frac{-1}{2^{M-1}}}$$
$$\geq 2T\, T^{\frac{-1}{2^{\log\log T}}} \qquad\qquad (M \geq \log\log T + 1)$$
$$\geq T.$$

$\qquad\square$

**Corollary A.7.** *Let $\widehat{\theta}_k$ be the estimator of $\theta^*$ calculated at the end of the $k^{th}$ batch. Then for any vector $x$ the following holds with probability greater than $1 - \frac{\log\log T}{T^2}$ for every batch $k \geq 2$.*

$$|\langle x, \theta^* - \widehat{\theta}_k \rangle| \leq \|x\|_{\mathbf{H}_k^{-1}}\, \gamma(\lambda)$$

---

[13]For logistic rewards, we note that instead of using the worst case bound of $\kappa$, one can make use of an upper bound on $S \coloneqq \|\theta\|$ as done in [20] That is, we lower bound $\dot{\mu}\left(\langle x, \theta^* \rangle\right) \geq \dot{\mu}\left(S\, \|x\|\right)$ and use $\mathbf{H}^* \succeq \sum_t \dot{\mu}\left(S\, \|x_t\|\right) x_t x_t^\intercal$.

*Proof.* This result is a direct consequence of Lemma A.2 and the definition of $\gamma(\lambda)$ (see 9). According to the lemma, we have

$$
\begin{aligned}
|\langle x, \theta^* - \widehat{\theta}_k \rangle| &\leq \|x\|_{\mathbf{H}_k^{*-1}} \left\| \theta^* - \widehat{\theta}_k \right\|_{\mathbf{H}_k^*} \\
&\leq \|x\|_{\mathbf{H}_k^{*-1}} \gamma(\lambda)
\end{aligned}
$$

Using Lemma A.5, we can further bound $\|x\|_{\mathbf{H}_k^{*-1}} \leq \|x\|_{\mathbf{H}_{k}^{-1}}$. Finally, a union bound over all batches and considering the fact that there are at most $\log \log T$ batches (Claim A.6) we establish the corollary's claim. $\qquad\square$

**Claim A.8.** *For any $x \in [0, M]$ the following holds*

$$
e^x \leq \left( e^M - 1 \right) \frac{x}{M} + 1.
$$

*Proof.* The claim follows from the convexity of $e^x$. $\qquad\square$

**Lemma A.9.** *Let $x \in \mathcal{X}$ be the selected in any round of batch $k \geq 2$ in the algorithm, and let $x^*$ be the optimal arm in the arm set $\mathcal{X}$, i.e., $x^* = \arg\max_{x \in \mathcal{X}} \mu\left(\langle x, \theta^* \rangle\right)$. With probability greater than $1 - \frac{\log \log T}{T^2}$, the following inequality holds-*

$$
\mu\left(\langle x^*, \theta^* \rangle\right) - \mu\left(\langle x, \theta^* \rangle\right) \leq 6\phi(\lambda) \sum_{\substack{y \in \{x, x^*\} \\ \widetilde{y} \in \{\widetilde{x}, \widetilde{x}^*\}}} \|y\|_{\mathbf{V}^{-1}} \|\widetilde{y}\|_{\mathbf{H}_{k-1}} + 2\gamma(\lambda) \sqrt{\dot{\mu}\left(\langle x^*, \theta^* \rangle\right)} \left( \|\widetilde{x}^*\|_{\mathbf{H}_{k-1}^{-1}} + \|\widetilde{x}\|_{\mathbf{H}_{k-1}^{-1}} \right)
$$

*Proof.* We begin by applying Taylor's theorem, which yields the following for some $z$ between $\langle x, \theta^* \rangle$ and $\langle x^*, \theta^* \rangle$

$$
\begin{aligned}
&|\mu\left(\langle x^*, \theta^* \rangle\right) - \mu\left(\langle x, \theta^* \rangle\right)| &&(14)\\
&= \dot{\mu}(z) \, |\langle x^*, \theta^* \rangle - \langle x, \theta^* \rangle| \\
&= \dot{\mu}(z) \left| \langle x^*, \theta^* \rangle - \langle x^*, \widehat{\theta}_{k-1} \rangle + \langle x^*, \widehat{\theta}_{k-1} \rangle - \langle x, \widehat{\theta}_{k-1} \rangle + \langle x, \widehat{\theta}_{k-1} \rangle - \langle x, \theta^* \rangle \right| \\
&\leq \dot{\mu}(z) \left( \left| \langle x^*, \theta^* \rangle - \langle x^*, \widehat{\theta}_{k-1} \rangle \right| + \left| \langle x^*, \widehat{\theta}_{k-1} \rangle - \langle x, \widehat{\theta}_{k-1} \rangle \right| + \left| \langle x, \widehat{\theta}_{k-1} \rangle - \langle x, \theta^* \rangle \right| \right) \\
&\leq 2\dot{\mu}(z) \left( \|x^*\|_{\mathbf{H}_{k-1}^{-1}} \gamma(\lambda) + \|x\|_{\mathbf{H}_{k-1}^{-1}} \gamma(\lambda) \right) &&\text{(via Corollary A.7)} \\
&\leq 2\dot{\mu}(z)\gamma(\lambda) \left( \sqrt{\frac{\beta(x^*)}{\dot{\mu}\left(\langle x^*, \widehat{\theta}_w \rangle\right)}} \|\widetilde{x}^*\|_{\mathbf{H}_{k-1}^{-1}} + \sqrt{\frac{\beta(x)}{\dot{\mu}\left(\langle x, \widehat{\theta}_w \rangle\right)}} \|\widetilde{x}\|_{\mathbf{H}_{k-1}^{-1}} \right) \\
&\leq 2\gamma(\lambda)\sqrt{\dot{\mu}(z)} \sqrt{\frac{\dot{\mu}(z)\beta(x^*)}{\dot{\mu}\left(\langle x^*, \widehat{\theta}_w \rangle\right)}} \|\widetilde{x}^*\|_{\mathbf{H}_{k-1}^{-1}} + 2\sqrt{\dot{\mu}(z)}\gamma(\lambda) \sqrt{\frac{\dot{\mu}(z)\beta(x)}{\dot{\mu}\left(\langle x, \widehat{\theta}_w \rangle\right)}} \|\widetilde{x}\|_{\mathbf{H}_{k-1}^{-1}} &&(15)
\end{aligned}
$$

We now invoke Lemmas C.2 and A.4 to obtain

$$\sqrt{\frac{\dot{\mu}(z)}{\dot{\mu}\left(\langle x, \widehat{\theta}_w \rangle\right)}}\beta(x) \leq \sqrt{\exp\left(\min\left\{2S, \left|z - \langle x, \widehat{\theta}_w \rangle\right|\right\}\right)\beta(x)}$$

$$\text{(by stated assumptions and Lemma C.2)}$$

$$\leq \exp\left(\min\left\{S, \frac{|\langle x, \theta^* \rangle - \langle x, \widehat{\theta}_w \rangle| + |\langle x, \theta^* \rangle - z|}{2}\right\}\right)\sqrt{\beta(x)}$$

$$\leq \exp\left(\min\left\{S, \frac{|\langle x, \theta^* \rangle - \langle x, \widehat{\theta}_w \rangle| + |\langle x, \theta^* \rangle - \langle x^*, \theta^* \rangle|}{2}\right\}\right)\sqrt{\beta(x)}$$

$$\text{(since } z \in [\langle x, \theta^* \rangle, \langle x_t^*, \theta^* \rangle])$$

$$\leq \exp\left(\min\left\{S, \frac{3\sqrt{\kappa}\,\|x\|_{\mathbf{V}^{-1}}\,\gamma(\lambda) + 2\sqrt{\kappa}\,\|x^*\|_{\mathbf{V}^{-1}}\,\gamma(\lambda)}{2}\right\}\right)\sqrt{\beta(x)}$$

$$\text{(using Lemma A.4 and the elimination criteria)}$$

$$\leq \exp\left(\min\left\{2S, 2\sqrt{\kappa}\,\|x\|_{\mathbf{V}^{-1}}\,\gamma(\lambda) + \sqrt{\kappa}\,\|x^*\|_{\mathbf{V}^{-1}}\,\gamma(\lambda)\right\}\right)$$

$$\text{(substituting the definition of } \beta(x))$$

Similarly, we also have

$$\sqrt{\dot{\mu}(z)} \leq \sqrt{\dot{\mu}\left(\langle x^*, \theta^* \rangle\right)}\exp\left(\min\left\{S, \sqrt{\kappa}\,\|x\|_{\mathbf{V}^{-1}}\,\gamma(\lambda) + \sqrt{\kappa}\,\|x^*\|_{\mathbf{V}^{-1}}\,\gamma(\lambda)\right\}\right).$$

Further, we can simplify each term in equation (15) as

$$\sqrt{\dot{\mu}(z)}\left(\sqrt{\frac{\dot{\mu}(z)\beta(x^*)}{\dot{\mu}\left(\langle x^*, \widehat{\theta}_w \rangle\right)}}\,\|\widetilde{x}^*\|_{\mathbf{H}_{k-1}^{-1}}\right)$$

$$\leq 2\sqrt{\dot{\mu}\left(\langle x^*, \theta^* \rangle\right)}\left(\|\widetilde{x}^*\|_{\mathbf{H}_{k-1}^{-1}}\exp\left(\min\left\{3S, 3\sqrt{\kappa}\gamma(\lambda)\left(\|x\|_{\mathbf{V}^{-1}} + \|x^*\|_{\mathbf{V}^{-1}}\right)\right\}\right)\right)$$

$$\leq 6\frac{\sqrt{\dot{\mu}\left(\langle x^*, \theta^* \rangle\right)}\,\kappa\,e^{3S}\,\gamma(\lambda)^2}{S}\left(\|x^*\|_{\mathbf{V}^{-1}} + \|x\|_{\mathbf{V}^{-1}}\right)\|\widetilde{x}^*\|_{\mathbf{H}_{k-1}^{-1}} + 2\gamma(\lambda)\sqrt{\dot{\mu}\left(\langle x^*, \theta^* \rangle\right)}\,\|\widetilde{x}^*\|_{\mathbf{H}_{k-1}^{-1}}$$

$$\text{(via Claim A.8)}$$

$$\leq 6\frac{\sqrt{\dot{\mu}\left(\langle x^*, \theta^* \rangle\right)}\,\kappa\,e^{3S}\,\gamma(\lambda)^2}{S}\left(\|x^*\|_{\mathbf{V}^{-1}} + \|x\|_{\mathbf{V}^{-1}}\right)\|\widetilde{x}^*\|_{\mathbf{H}_{k-1}^{-1}}$$

$$+ 2\gamma(\lambda)\sqrt{\dot{\mu}\left(\langle x^*, \theta^* \rangle\right)}\,\|\widetilde{x}^*\|_{\mathbf{H}_{k-1}^{-1}}$$

$$\leq 6\sqrt{\dot{\mu}\left(\langle x^*, \theta^* \rangle\right)}\phi(\lambda)\left(\|x^*\|_{\mathbf{V}^{-1}} + \|x\|_{\mathbf{V}^{-1}}\right)\|\widetilde{x}^*\|_{\mathbf{H}_{k-1}^{-1}} + 2\gamma(\lambda)\sqrt{\dot{\mu}\left(\langle x^*, \theta^* \rangle\right)}\,\|\widetilde{x}^*\|_{\mathbf{H}_{k-1}^{-1}}$$

Finally, we substitute the above bound in (15) to obtain

$$|\mu\left(\langle x^*, \theta^* \rangle\right) - \mu\left(\langle x, \theta^* \rangle\right)| \leq 6\sqrt{\dot{\mu}\left(\langle x^*, \theta^* \rangle\right)}\phi(\lambda)\left(\|x^*\|_{\mathbf{V}^{-1}} + \|x\|_{\mathbf{V}^{-1}}\right)\left(\|\widetilde{x}^*\|_{\mathbf{H}_{k-1}^{-1}} + \|\widetilde{x}\|_{\mathbf{H}_{k-1}^{-1}}\right)$$

$$+ 2\sqrt{\dot{\mu}\left(\langle x^*, \theta^* \rangle\right)}\left(\|\widetilde{x}^*\|_{\mathbf{H}_{k-1}^{-1}} + \|\widetilde{x}\|_{\mathbf{H}_{k-1}^{-1}}\right)$$

$$\square$$

For Phase $k$, the distribution of the remaining arms after the elimination step ($\mathcal{X}$ in line 10 of the Algorithm 1) is represented as $\mathcal{D}_k$.

**Lemma A.10.** *During any round in batch $k$ of Algorithm 1, and for an absolute constant $c$, we have*

$$\mathbb{E}\left[|\mu\left(\langle x^*, \theta^* \rangle\right) - \mu\left(\langle x, \theta^* \rangle\right)|\right] \leq c\left(\frac{\phi(\lambda)d^2}{\sqrt{\tau_1\,\tau_{k-1}}} + \frac{\gamma(\lambda)}{\sqrt{\tau_{k-1}}}\left(\frac{d}{\sqrt{\widehat{\kappa}}} \wedge \sqrt{\frac{d\log d}{\kappa^*}}\right)\right)$$

*Proof.* The proof here invokes Lemma A.9. We begin by noting that

$$\mathop{\mathbb{E}}_{\mathcal{X}\sim\mathcal{D}_k}\sum_{\substack{y\in\{x,x^*\}\\\widetilde{y}\in\{\widetilde{x},\widetilde{x}^*\}}}\|y\|_{\mathbf{V}^{-1}}\|\widetilde{y}\|_{\mathbf{H}_{k-1}^{-1}}\leq 4\mathop{\mathbb{E}}_{\mathcal{X}\sim\mathcal{D}_k}\left[\max_{x\in\mathcal{X}}\|x\|_{\mathbf{V}^{-1}}\max_{\mathbf{x}\in\mathcal{X}}\|\widetilde{x}\|_{\mathbf{H}_{k-1}^{-1}}\right]$$

$$\leq 4\sqrt{\mathop{\mathbb{E}}_{\mathcal{X}\sim\mathcal{D}_k}\left[\max_{x\in\mathcal{X}}\|x\|_{\mathbf{V}^{-1}}^2\right]\mathop{\mathbb{E}}_{\mathcal{X}\sim\mathcal{D}_k}\left[\max_{\mathbf{x}\in\mathcal{X}}\|\widetilde{x}\|_{\mathbf{H}_{k-1}^{-1}}^2\right]}$$
$$\text{(via Jensen's inequality)}$$

$$\leq 4\sqrt{\mathop{\mathbb{E}}_{\mathcal{X}\sim\mathcal{D}}\left[\max_{x\in\mathcal{X}}\|x\|_{\mathbf{V}^{-1}}^2\right]\mathop{\mathbb{E}}_{\mathcal{X}\sim\mathcal{D}_{k-1}}\left[\max_{\mathbf{x}\in\mathcal{X}}\|\widetilde{x}\|_{\mathbf{H}_{k-1}^{-1}}^2\right]}$$
$$\text{(via Claim A.11)}$$

$$\leq c\left(\sqrt{\frac{d^2}{\tau_1}\cdot\frac{d^2}{\tau_{k-1}}}\right)\qquad\text{(using Lemma A.17)}$$

We also have

$$\mathop{\mathbb{E}}_{\mathcal{X}\sim\mathcal{D}_k}\left[\sqrt{\dot{\mu}\left(\langle x^*,\theta^*\rangle\right)}\left(\|\widetilde{x}^*\|_{\mathbf{H}_{k-1}^{-1}}+\|\widetilde{x}\|_{\mathbf{H}_{k-1}^{-1}}\right)\right]$$

$$\leq 2\mathop{\mathbb{E}}_{\mathcal{X}\sim\mathcal{D}_k}\left[\sqrt{\dot{\mu}\left(\langle x^*,\theta^*\rangle\right)}\max_{x\in\mathcal{X}}\|x\|_{\mathbf{H}_{k-1}^{-1}}\right]$$

$$\leq 2\min\left\{\sqrt{\kappa^*}\mathop{\mathbb{E}}_{\mathcal{X}\sim\mathcal{D}_k}\left[\max_{x\in\mathcal{X}}\|x\|_{\mathbf{H}_{k-1}^{-1}}\right],\sqrt{\mathop{\mathbb{E}}_{\mathcal{X}\sim\mathcal{D}_k}\left[\dot{\mu}\left(\langle x^*,\theta^*\rangle\right)\right]\mathop{\mathbb{E}}_{\mathcal{X}\sim\mathcal{D}_k}\left[\|x\|_{\mathbf{H}_{k-1}^{-1}}^2\right]}\right\}$$
$$\text{(using the definition of }\kappa^*\text{ for the first bound and Jensen for the second)}$$

$$\leq c\left(\sqrt{\frac{d\log d}{\kappa^*\tau_{k-1}}}\wedge\sqrt{\frac{d^2}{\widehat{\kappa}\tau_{k-1}}}\right)\qquad\text{(using Lemma A.17)}$$

Substituting the above bounds in Lemma A.9 we obtained the stated inequality. This completes the proof. $\qquad\square$

## A.4   Proof of Theorem 3.2

We trivially upper bound the regret incurred during the warm-up batch as $\tau_1 R$; recall that $R$ denotes the upper bound on the rewards and $\tau_1$ denotes the length of the first (warm-up) batch; see equation (5)).

For each batch $k$ and an absolute constant $c$, Lemma A.10 gives us

$$\mathbb{E}\left[\sum_{t\in\mathcal{T}_k}\mu\left(\langle x_t^*,\theta^*\rangle\right)-\mu\left(\langle x_t,\theta^*\rangle\right)\right]\leq\tau_k\cdot c\left(\frac{\phi(\lambda)d^2}{\sqrt{\tau_1\,\tau_{k-1}}}+\frac{\gamma(\lambda)}{\sqrt{\tau_{k-1}}}\left(\frac{d}{\sqrt{\widehat{\kappa}}}\wedge\sqrt{\frac{d\log d}{\kappa^*}}\right)\right)$$

$$\leq c\left(\frac{\phi(\lambda)d^2}{\sqrt{\tau_1}}\alpha+\left(\frac{d}{\sqrt{\widehat{\kappa}}}\wedge\sqrt{\frac{d\log d}{\kappa^*}}\right)\gamma(\lambda)\alpha\right)\quad\text{(via (3))}$$

Since there are at most $\log\log T$ batches, we can upper bound the regret as

$$R_T\leq c\left(\tau_1 R+\frac{\phi(\lambda)d^2}{\sqrt{\tau_1}}\alpha+\left(\frac{d}{\sqrt{\widehat{\kappa}}}\wedge\sqrt{\frac{d\log d}{\kappa^*}}\right)\gamma(\lambda)\alpha\right)\log\log(T)$$

Setting $\tau_1=\left(\frac{\phi(\lambda)d^2\alpha}{R}\right)^{2/3}$ we get

$$R_T\leq O\left(\left(\frac{\phi(\lambda)d^2\alpha}{R}\right)^{2/3}+\left(\frac{d}{\sqrt{\widehat{\kappa}}}\wedge\sqrt{\frac{d\log d}{\kappa^*}}\right)\gamma(\lambda)\alpha\right)\log\log(T)$$

Now with the choice of $\lambda=20dR\log T$, we have

$$\gamma(\lambda)\leq\gamma\quad\text{where }\gamma=30RS\sqrt{d\log T}.$$

Substituting $\alpha = T^{\frac{1}{2(1-2^{-M})}}$ and $\phi(\lambda) = \frac{\sqrt{\kappa}\,e^{3S}\,\gamma^2}{S}$ we get

$$R_T \leq O\left(\left(\sqrt{\kappa}d^3\,e^{3S}\,RST^{\frac{1}{2(1-2^{-M})}}\log T\right)^{2/3} + \left(\frac{d}{\sqrt{\widehat{\kappa}}} \wedge \sqrt{\frac{d\log d}{\kappa^*}}\right) RST^{\frac{1}{2(1-2^{-M})}}\sqrt{d\log T}\right).$$

## A.5 Optimal Design Guarantees

In this section, we study the optimal design policies utilized in different batches of the algorithm. Specifically, $\pi_G$ denotes the G-OPTIMAL DESIGN policy applied during the warm-up batch, while $\pi_k$ refers to the DISTRIBUTIONAL OPTIMAL DESIGN policy calculated at the end of batch $k$ ( and used in the $(k+1)^{th}$ batch). Recall that the distribution of the remaining arms after the elimination step ( $\mathcal{X}$ in line 10 of the Algorithm) is represented as $\mathcal{D}_k$. We define expected design matrices for each policy:

$$\mathbf{W}_G \coloneqq \underset{\mathcal{X}\sim\mathcal{D}}{\mathbb{E}}\left[\underset{x\sim\pi_G(\mathcal{X})}{\mathbb{E}}\left[xx^\mathsf{T}|\mathcal{X}\right]\right]$$

$$\mathbf{W}_k \coloneqq \underset{\mathcal{X}\sim\mathcal{D}_k}{\mathbb{E}}\left[\underset{x\sim\pi_k(\mathcal{X})}{\mathbb{E}}\left[\widetilde{x}\widetilde{x}^\mathsf{T}|\mathcal{X}\right]\right]$$

Recall, for all batches starting from the second batch ($k \geq 2$), we employ the scaled arm set, denoted as $\widetilde{\mathcal{X}}$, for learning and action selection under the DISTRIBUTIONAL OPTIMAL DESIGN policy. However, during the initial warm-up batch, we utilize the original, unscaled arm set.

**Claim A.11.** *The following holds for any positive semidefinite matrix* $\mathbf{A}$ *and any batch* $k$-

$$\underset{\mathcal{X}\sim\mathcal{D}_k}{\mathbb{E}}\max_{x\in\mathcal{X}}\|x\|_A \leq \underset{\mathcal{X}\sim\mathcal{D}_j}{\mathbb{E}}\max_{x\in\mathcal{X}}\|x\|_A \quad \forall j \in [k-1].$$

This is due to the fact that the set of surviving arms in batch $k$ is always a smaller set than the previous batches.

**Lemma A.12** (Lemma 4 [23]). *The expected data matrix* $\mathbf{W}_G$ *satisfies. We have*

$$\underset{\mathcal{X}\sim\mathcal{D}}{\mathbb{E}}\left[\max_{x\in\mathcal{X}}\|x\|^2_{\mathbf{W}_G^{-1}}\right] \leq d^2$$

**Lemma A.13** (Theorem 5 [23]). *Let the* DISTRIBUTIONAL OPTIMAL DESIGN $\pi$ *which has been learnt from* $s$ *independent samples* $\mathcal{X}_1,\ldots\mathcal{X}_s \sim \mathcal{D}$ *and let* $\mathbf{W}$ *denote the expected data matrix,* $\mathbf{W} = \mathbb{E}_{\mathcal{X}\sim\mathcal{D}}\left[\mathbb{E}_{x\sim\pi(\mathcal{X})}\left[xx^\mathsf{T}|\mathcal{X}\right]\right]$. *We have*

$$\mathbb{P}\left\{\underset{\mathcal{X}\sim\mathcal{D}}{\mathbb{E}}\left[\max_{x\in\mathcal{X}}\|x\|_{\mathbf{W}^{-1}}\right] \leq O\left(\sqrt{d\log d}\right)\right\} \geq 1 - \exp\left(O\left(d^4\log^2 d\right) - sd^{-12}\cdot 2^{-16}\right). \quad (16)$$

**Lemma A.14.** *Under the notation of Lemma A.13, we have*

$$\underset{\mathcal{X}\sim\mathcal{D}}{\mathbb{E}}\left[\max_{x\in\mathcal{X}}\|x\|^2_{\mathbf{W}^{-1}}\right] \leq 2d^2. \quad (17)$$

*Proof.* Recall that the DISTRIBUTIONAL OPTIMAL DESIGN policy samples according to the $\pi_G$ policy with half probability. Hence, we have

$$\mathbf{W} = \underset{\mathcal{X}\sim\mathcal{D}_k}{\mathbb{E}}\left[\underset{x\sim\pi(\mathcal{X})}{\mathbb{E}}\left[xx^\mathsf{T}|\mathcal{X}\right]\right] \succeq \underset{\mathcal{X}\sim\mathcal{D}_k}{\mathbb{E}}\left[\frac{1}{2}\underset{x\sim\pi_G(\mathcal{X})}{\mathbb{E}}\left[xx^\mathsf{T}|\mathcal{X}\right]\right] = \frac{1}{2}\mathbf{W}_G$$

Therefore,

$$\underset{\mathcal{X}\sim\mathcal{D}}{\mathbb{E}}\left[\max_{x\in\mathcal{X}}\|x\|^2_{\mathbf{W}^{-1}}\right] \leq 2\underset{\mathcal{X}\sim\mathcal{D}}{\mathbb{E}}\left[\max_{x\in\mathcal{X}}\|x\|^2_{\mathbf{W}_G^{-1}}\right] \leq 2d^2.$$

$\square$

**Lemma A.15** (Matrix Chernoff [27, 23]). *Let* $x_1, x_3, \ldots, x_n \sim \mathcal{D}$ *be vectors, with* $\|x_t\| \leq 1$, *then we have*

$$\mathbb{P}\left\{3\varepsilon n\mathbf{I} + \sum_{i=1}^n x_i x_i^\mathsf{T} \succeq \frac{n}{8}\underset{x\sim\mathcal{D}}{\mathbb{E}}\left[xx^\mathsf{T}\right]\right\} \geq 1 - 2d\exp\left(-\frac{\varepsilon n}{8}\right)$$

**Corollary A.16.** *In Algorithm 1 the warm-up matrix* $\mathbf{V}$, *with* $\lambda \geq 16 \log(Td)$, *satisfies the following with probability greater than* $1 - \frac{1}{T^2}$.

$$\mathbf{V} \succeq \frac{\tau_1}{8} \mathop{\mathbb{E}}_{\mathcal{X} \sim \mathcal{D}} \mathop{\mathbb{E}}_{x \sim \pi_G(\mathcal{X})} xx^{\mathsf{T}}$$

*Similarly* $\mathbf{H}_k$, *with* $\lambda \geq 6 \log(Td)$ *satisfies the following for each batch* $k \geq 2$ *with probability greater than* $1 - \frac{1}{T^2}$

$$\mathbf{H}_k \succeq \frac{\tau_k}{8} \mathop{\mathbb{E}}_{\mathcal{X} \sim \mathcal{D}_k} \mathop{\mathbb{E}}_{\widetilde{x} \sim \pi_k(\mathcal{X})} \widetilde{x}\widetilde{x}^{\mathsf{T}}.$$

*Proof.* The results for both $\mathbf{V}$ and $\mathbf{H}_k$ are obtained directly by applying Lemma A.15 with $\varepsilon = \frac{\log(T)}{\tau_1}$ and $\varepsilon = \frac{\log(T)}{\tau_k}$, respectively. $\qquad\square$

We note that the analysis of [23] gives an optimal guarantee (in expectation) on $\|x\|_{\mathbf{V}^{-1}}$, but not on $\|x\|^2_{\mathbf{V}^{-1}}$. We obtain such a bound here and use it in the analysis.

**Lemma A.17.** *The following holds with probability greater than* $1 - \frac{1}{T^2}$

$$\mathop{\mathbb{E}}_{\mathcal{X} \sim \mathcal{D}} \max_{x \in \mathcal{X}} \|x\|^2_{\mathbf{V}^{-1}} \leq O\left(\frac{d^2}{\tau_1}\right) \tag{18}$$

$$\mathop{\mathbb{E}}_{\mathcal{X} \sim \mathcal{D}_k} \max_{x \in \mathcal{X}} \|\widetilde{x}\|^2_{\mathbf{H}_k^{-1}} \leq O\left(\frac{d^2}{\tau_k}\right) \quad \forall k \in [M] \tag{19}$$

*We also have that for sufficiently large* $T \gtrsim O\left(d^{32}(\log 2T)^2\right)$, *the following holds with probability greater than* $1 - \frac{\log\log T}{T}$

$$\mathop{\mathbb{E}}_{\mathcal{X} \sim \mathcal{D}_k} \max_{x \in \mathcal{X}} \|\widetilde{x}\|_{\mathbf{H}_k^{-1}} \leq O\left(\sqrt{\frac{d}{\tau_k}}\right) \quad \forall k \in [M] \tag{20}$$

*Proof.* First, we note from Corollary A.16 that the following holds with high probability

$$\|x\|_{\mathbf{V}^{-1}} \leq \frac{8}{\tau_1} \|x\|_{\mathbf{W}_G^{-1}}$$

$$\|\widetilde{x}\|_{\mathbf{H}_k^{-1}} \geq \frac{8}{\tau_1} \|\widetilde{x}\|_{\mathbf{W}_k^{-1}}$$

We obtain the first two inequalities, (18) and (19), by a direct use of Corollary A.16. For (20) we note that for every phase we have at least $O(\sqrt{T})$ samples for learning the DISTRIBUTIONAL OPTIMAL DESIGN policy (for any M). Since, $T \geq d^{32} \log 2T^2$ the event stated in Lemma A.13 holds with probability greater than $1 - \frac{1}{T^2}$. $\qquad\square$

# B  Regret Analysis of RS-GLinCB

Recall $\mathcal{T}_o$ denotes the set of rounds when switching criterion I is satisfied. We write $\tau_o$ to denote the size of the set $\tau_o = |\mathcal{T}_o|$. We define the following (scaled) data matrix

$$\mathbf{H}_w^* = \sum_{s \in \mathcal{T}_o} \dot{\mu}\left(\langle x_s, \theta^* \rangle\right) x_s x_s^{\mathsf{T}} + \lambda \mathbf{I}.$$

We will specify the regularizer $\lambda$ later. We also define

$$\gamma := \mathbf{c}RS\sqrt{d \log \frac{T}{\delta}}$$

Below, we state the main concentration bound used in the proof.

**Lemma B.1** (Theorem 1 of [6])**.** *Let $\{\mathcal{F}_t\}_{t=1}^{\infty}$ be a filtration. Let $\{x_t\}_{t=1}^{\infty}$ be a stochastic process in $B_1(d)$ such that $x_t$ is $\mathcal{F}_t$ measurable. Let $\{\eta_t\}_{t=1}^{\infty}$ be a martingale difference sequence such that $\eta_t$ is $\mathcal{F}_t$ measurable. Furthermore, assume we have $|\eta_t| \leq 1$ almost surely, and denote $\sigma_t^2 = \mathbb{V}[\eta_t|\mathcal{F}_t]$. Let $\lambda > 0$ and for any $t \geq 1$ define:*

$$S_t = \sum_{s=1}^{t-1} \eta_s x_s \qquad \mathbf{H}_t = \sum_{s=1}^{t-1} \sigma_s^2 x_s x_s^\intercal + \lambda \mathbf{I}$$

*Then, for any $\delta \in (0,1]$,*

$$\mathbb{P}\left[\exists t \geq 1 : \|S_t\|_{\mathbf{H}_t^{-1}} \geq \frac{\sqrt{\lambda}}{2} + \frac{2}{\sqrt{\lambda}}\log\left(\frac{\det(\mathbf{H}_t)^{\frac{1}{2}}}{\lambda^{d/2}\delta}\right) + \frac{2}{\sqrt{\lambda}}d\log(2)\right] \leq \delta$$

## B.1 Confidence Sets for Switching Criterion I rounds

**Lemma B.2.** *At any round $t$, let $\widehat{\theta}_o$ be the maximum likelihood estimate calculated using set of rewards observed in the rounds $\mathcal{T}_o$. With probability at least $1 - \delta$ we have*

$$\|\widehat{\theta}_o - \theta^*\|_{\mathbf{H}_w^*} \leq \gamma.$$

*Proof.* Let us define the matrix $\mathbf{G}_w = \sum_{s \in \mathcal{T}_o} \alpha(x, \theta^*, \widehat{\theta}_o) x_s x_s^\intercal + \lambda \mathbf{I}$. First, we note that by self-concordance property of $\mu$ (Lemma C.2), $\mathbf{G}_w \succeq \frac{1}{1+2RS}\mathbf{H}_w^*$. Hence,

$$\begin{aligned}
\|\widehat{\theta}_o - \theta^*\|_{\mathbf{H}_w^*} &\leq \sqrt{1+2RS}\|\widehat{\theta}_o - \theta^*\|_{\mathbf{G}_w} \\
&= \sqrt{1+2RS}\left\|\mathbf{G}_w\left(\widehat{\theta}_o - \theta^*\right)\right\|_{\mathbf{G}_w^{-1}} \\
&= \sqrt{1+2RS}\left\|\sum_{s \in \mathcal{T}_o}\left(\langle\widehat{\theta}_o, x_s\rangle - \langle\theta^*, x_s\rangle\right)\alpha(x_s, \widehat{\theta}_o, \theta^*)x_s + \lambda\widehat{\theta}_o - \lambda_t\theta^*\right\|_{\mathbf{G}_w^{-1}} \\
&= \sqrt{1+2RS}\left\|\sum_{s \in \mathcal{T}_o}\left(\mu\left(\langle\widehat{\theta}_o, x_s\rangle\right) - \mu\left(\langle\theta^*, x_s\rangle\right)\right)x_s + \lambda\widehat{\theta}_o - \lambda_t\theta^*\right\|_{\mathbf{G}_w^{-1}} \\
&\qquad\qquad\qquad\qquad\qquad\qquad\qquad\qquad\qquad\qquad\qquad\text{(Taylor's theorem)} \\
&\leq (1+2RS)\left\|\sum_{s \in \mathcal{T}_o}\left(\mu\left(\langle\widehat{\theta}_o, x_s\rangle\right) - \mu\left(\langle\theta^*, x_s\rangle\right)\right)x_s + \lambda\widehat{\theta}_o - \lambda\theta^*\right\|_{\mathbf{H}_w^{*-1}} \\
&\qquad\qquad\qquad\qquad\qquad\qquad\qquad\qquad\qquad\qquad\quad(\mathbf{G}_w \succeq \frac{1}{1+2RS}\mathbf{H}_w^*)
\end{aligned}$$

Since $\widehat{\theta}_o$ is the maximum likelihood estimate, by optimality condition, we have the following relation: $\sum_{s \in \mathcal{T}_o}\mu\left(\langle x_s, \widehat{\theta}_o\rangle\right)x_s + \lambda\widehat{\theta}_o = \sum_{s \in \mathcal{T}_o}r_s x_s$. Substituting this above, we get

$$\begin{aligned}
\|\widehat{\theta}_o - \theta^*\|_{\mathbf{H}_w^*} &\leq (1+2RS)\left\|\sum_{s \in \mathcal{T}_o}\left(r_s - \mu\left(\langle\theta^*, x_s\rangle\right)\right)x_s - \lambda\theta^*\right\|_{\mathbf{H}_w^{*-1}} \\
&\leq (1+2RS)\left\|\sum_{s \in \mathcal{T}_o}\left(r_s - \mu\left(\langle\theta^*, x_s\rangle\right)\right)x_s\right\|_{\mathbf{H}_w^{*-1}} + \lambda\|\theta^*\|_{\mathbf{H}_w^{*-1}} \\
&\leq (1+2RS)\left\|\sum_{s \in \mathcal{T}_o}\eta_s x_s\right\|_{\mathbf{H}_w^{*-1}} + S\sqrt{\lambda}. \qquad (\mathbf{H}_w^* \succeq \lambda_t\mathbf{I}, \|\theta^*\|_2 \leq S)
\end{aligned}$$

where $\eta_s := \left(r_s - \mu\left(\langle\theta^*, x_s\rangle\right)\right)$.

We will now apply Lemma B.1 $\eta_s$ scaled by $R$. First note that $\left\|\sum_{s \in \mathcal{T}_o}\eta_s x_s\right\|_{(\mathbf{H}_w^*)^{-1}} = \left\|\sum_{s \in \mathcal{T}_o}\frac{\eta_s}{R}x_s\right\|_{(R^2\mathbf{H}_w^*)^{-1}}$ which, in turn ensures that the noise variable is upper bounded by 1.

Applying B.1 we get

$$\|\widehat{\theta}_o - \theta^*\|_{\mathbf{H}_w^*} \le S\sqrt{R^2\lambda}$$
$$+ (1 + 2RS)\left(\frac{\sqrt{R^2\lambda}}{2} + \frac{2d}{\sqrt{R^2\lambda}}\log\left(1 + \frac{\tau_o}{R^2\lambda d}\right) + \frac{2}{\sqrt{R^2\lambda}}\log\left(\frac{1}{\delta}\right) + \frac{2}{\sqrt{R^2\lambda}}d\log(2)\right)$$

Simplifying constants and setting $\sqrt{R^2\lambda} = \mathbf{c}\sqrt{d\log(T) + \log(1/\delta)}$, we have $\left\|\widehat{\theta}_o - \theta^*\right\|_{\mathbf{H}_w^*} \le$
$\mathbf{c}RS\sqrt{d\log(T/\delta) + \log(1/\delta)} \le \mathbf{c}RS\sqrt{d\log(T/\delta)}$. $\qquad\square$

## B.2 Confidence Sets for non-Switching Criterion I rounds

We define $\mathcal{E}_w$ be the event defined in Lemma B.2, that is, $\mathcal{E}_w = \{\|\widehat{\theta}_o - \theta^*\|_{\mathbf{H}_w^*} \le \gamma\}$.

**Lemma B.3.** *If in round $t$ the switching criteria I is not satisfied and the event $\mathcal{E}_w$ holds, we have*

$$|\langle x, \widehat{\theta}_o - \theta^*\rangle| \le \frac{1}{R} \quad \text{for all } x \in \mathcal{X}_t.$$

*Proof.*

$$
\begin{aligned}
|\langle x, \widehat{\theta}_o - \theta^*\rangle| &\le \|x\|_{\mathbf{H}_w^{*-1}} \cdot \|\widehat{\theta}_o - \theta^*\|_{\mathbf{H}_w^*} && \text{(Cauchy-Schwarz)} \\
&\le \|x\|_{\mathbf{H}_w^{*-1}}\gamma && \text{(via Lemma B.2)} \\
&\le \gamma\sqrt{\kappa}\|x\|_{\mathbf{V}_w^{-1}} && (\mathbf{V}_w \preceq \kappa\mathbf{H}_w^*) \\
&\le \gamma\sqrt{\kappa}\frac{1}{\sqrt{R^2\kappa\gamma^2}} && \text{(warm-up criteria is not satisfied)} \\
&\le \frac{1}{R}
\end{aligned}
$$

$\square$

Recall that $\mathbf{H}_t$ is defined in line 14 of Algorithm 2. Further, we define

$$\mathbf{H}_t^* = \sum_{s \in [t-1]\setminus\mathcal{T}_o} \dot{\mu}\left(\langle x_s, \theta^*\rangle\right) x_s x_s^{\mathsf{T}} + \lambda\mathbf{I}$$

.

**Corollary B.4.** *Under event $\mathcal{E}_w$, $\mathbf{H}_t \preceq \mathbf{H}_t^* \preceq e^2\mathbf{H}_t$*

*Proof.* For a given $s \in [t-1] \setminus \mathcal{T}_o$, let $\widehat{\theta}_o^s$ denote the value of $\widehat{\theta}_o$ in that round. Then, for all $x \in \mathcal{X}_s$, by Lemma C.2,

$$\dot{\mu}\left(\langle x, \widehat{\theta}_o^s\rangle\right)\exp(-R|\langle x, \widehat{\theta}_o^s - \theta^*\rangle|) \le \dot{\mu}\left(\langle x, \theta^*\rangle\right) \le \dot{\mu}\left(\langle x, \widehat{\theta}_o^s\rangle\right)\exp(R|\langle x, \widehat{\theta}_o^s - \theta^*\rangle|)$$

Applying lemma B.3, gives $e^{-1}\dot{\mu}\left(\langle x, \widehat{\theta}_o^s\rangle\right) \le \dot{\mu}\left(\langle x, \theta^*\rangle\right) \le e^1\dot{\mu}\left(\langle x, \widehat{\theta}_o^s\rangle\right)$. Thus,

$$\mathbf{H}_t = \sum_{s \in [t-1]\setminus\mathcal{T}_o} e^{-1}\dot{\mu}\left(\langle x_s, \widehat{\theta}_o^s\rangle\right) x_s x_s^{\mathsf{T}} + \lambda\mathbf{I} \preceq \sum_{s \in [t-1]\setminus\mathcal{T}_o} \dot{\mu}\left(\langle x_s, \theta^*\rangle\right) x_s x_s^{\mathsf{T}} + \lambda\mathbf{I} = \mathbf{H}_t^*$$

. Further, $\mathbf{H}_t^* \preceq \sum_{s \in [t-1]\setminus\mathcal{T}_o} e^2\frac{\dot{\mu}\left(\langle x_s, \widehat{\theta}_o^s\rangle\right)}{e} x_s x_s^{\mathsf{T}} + e^2\lambda\mathbf{I} = e^2\mathbf{H}_t$. $\qquad\square$

Recall that $\tau$ is the round when the Switching Criterion II (Line 9) is satisfied. Now we define the following quantities:

$$g_\tau(\theta) = \sum_{s \in [\tau-1] \setminus \mathcal{T}_o} \mu\left(\langle x_s, \theta \rangle\right) x_s + \lambda_\tau \theta$$

$$\mathbf{H}_\tau(\theta) = \sum_{s \in [\tau-1] \setminus \mathcal{T}_o} \dot{\mu}\left(\langle x_s, \theta \rangle\right) x_s x_s^{\mathsf{T}} + \lambda \mathbf{I}$$

$$\Theta = \left\{ \theta : \left\| \theta - \widehat{\theta}_o \right\|_{\mathbf{V}} \leq \gamma \sqrt{\kappa} \right\}$$

$$\tilde{\theta} = \arg\min_{\theta \in \mathbb{R}^d} \sum_{s \in [t-1] \setminus \mathcal{T}_o} \ell(\theta, x_s, r_s)$$

$$\beta := \mathbf{c}\sqrt{d \log(T/\delta)}$$

Moreover, recall the following definition $\widehat{\theta}_\tau$:

$$\widehat{\theta}_\tau = \arg\min_{\theta \in \Theta} \left\| g_\tau(\theta) - g_\tau(\tilde{\theta}) \right\|_{\mathbf{H}_\tau(\theta)^{-1}} \tag{21}$$

**Lemma B.5.** *Under event $\mathcal{E}_w$, $\left\| \widehat{\theta}_\tau - \theta^* \right\|_{\mathbf{V}} \leq 2\gamma\sqrt{\kappa}$*

*Proof.* First, we observe from Lemma B.2 that $\left\| \widehat{\theta}_o - \theta^* \right\|_{\mathbf{H}_w^*} \leq \gamma$. Using $\mathbf{V} \preceq \kappa \mathbf{H}_w^*$, we can write $\left\| \widehat{\theta}_o - \theta^* \right\|_{\mathbf{V}} \leq \gamma\sqrt{\kappa}$. This implies that $\theta^* \in \Theta$. Now, $\widehat{\theta}_\tau \in \Theta$ by virtue of being a feasible solution to the optimization in (21). Thus,

$$\begin{aligned} \left\| \widehat{\theta}_\tau - \theta^* \right\|_{\mathbf{V}} &= \left\| \widehat{\theta}_\tau - \widehat{\theta}_o + \widehat{\theta}_o - \theta^* \right\|_{\mathbf{V}} \\ &\leq \left\| \widehat{\theta}_\tau - \widehat{\theta}_o \right\|_{\mathbf{V}} + \left\| \widehat{\theta}_o - \theta^* \right\|_{\mathbf{V}} && \text{(triangle inequality)} \\ &\leq 2\gamma\sqrt{\kappa} \end{aligned}$$

$\square$

**Lemma B.6.** *Let $\delta \in (0,1)$. Then, under event $\mathcal{E}_w$, with probability $1-\delta$, $\left\| \widehat{\theta}_\tau - \theta^* \right\|_{\mathbf{H}_\tau^*} \leq \beta$.*

*Proof.* We have for all rounds $s \in [\tau-1] \setminus \mathcal{T}_o$,

$$\begin{aligned} |\langle x_s, \theta^* - \widehat{\theta}_\tau \rangle| &\leq \| x_s \|_{\mathbf{V}^{-1}} \left\| \theta^* - \widehat{\theta}_\tau \right\|_{\mathbf{V}} && \text{(Cauchy-Schwarz)} \\ &\leq \| x_s \|_{\mathbf{V}^{-1}} 2\gamma\sqrt{\kappa} && \text{(by Lemma B.5)} \\ &\leq 2\gamma\sqrt{\kappa} \frac{1}{R\gamma\sqrt{\kappa}} && \text{(warm up criterion not satisfied)} \\ &= \frac{2}{R} \end{aligned}$$

Also note that $\theta^* \in \Theta$. Hence,

$$\begin{aligned} \left\| \widehat{\theta}_\tau - \theta^* \right\|_{\mathbf{H}_\tau^*} &\leq 2(1+2) \left\| g_\tau(\theta^*) - \sum_{s \in [\tau-1]} r_s x_s \right\|_{\mathbf{H}^{*-1}} && \text{(by Lemma E.1)} \\ &\leq 6\mathbf{c}\sqrt{d \log(T/\delta)} && \text{(by Lemma B.1)} \end{aligned}$$

$\square$

Let the event in lemma B.6 be denoted by $\mathcal{E}_\tau$, or in other words, $\mathcal{E}_\tau = \{\|\widehat{\theta}_\tau - \theta^*\|_{\mathbf{H}_\tau^*} \leq \beta\}$.

## B.3 Bounding the instantaneous regret

In this subsection, we will only consider rounds $t \in [T]$ which does not satisfy Switching Criterion I. Let $x_t \in \mathcal{X}_t$ be the played arm defined via line 13 Algorithm 2. Further, let $x_t^* \in \mathcal{X}_t$ be the best available arm in that round.

**Corollary B.7.** *Under the event $\mathcal{E}_\tau$, for all $x \in \mathcal{X}_t$, we have, $|\langle x, \widehat{\theta}_\tau - \theta^* \rangle| \leq \beta \|x\|_{\mathbf{H}_\tau^{-1}}$ .*

*Proof.*

$$
\begin{aligned}
|\langle x, \widehat{\theta}_\tau - \theta^* \rangle| &\leq \|x\|_{\mathbf{H}_\tau^{*-1}} \cdot \|\widehat{\theta}_\tau - \theta^*\|_{\mathbf{H}_\tau^*} && \text{(Cauchy-Schwartz)} \\
&\leq \beta \|x\|_{\mathbf{H}_\tau^{*-1}} && \text{(by Lemma B.6)} \\
&\leq \beta \|x\|_{\mathbf{H}_\tau^{-1}} && (\mathbf{H}_\tau \preceq \mathbf{H}_\tau^*)
\end{aligned}
$$

$\square$

**Lemma B.8.** *Under event $\mathcal{E}_\tau$, $\langle x_t^* - x_t, \theta^* \rangle \leq 2\sqrt{2}\beta \|x_t\|_{\mathbf{H}_t^{*-1}}$ .*

*Proof.*

$$
\begin{aligned}
\langle x_t^*, \theta^* \rangle - \langle x_t, \theta^* \rangle &\leq \left( \langle x_t^*, \widehat{\theta}_\tau \rangle + \beta \|x_t^*\|_{\mathbf{H}_t^{*-1}} \right) - \left( \langle x_t, \widehat{\theta}_\tau \rangle - \beta \|x_t\|_{\mathbf{H}_t^{*-1}} \right) && \text{(by Corollary B.7)} \\
&\leq \left( \langle x_t, \widehat{\theta}_\tau \rangle + \beta \|x_t\|_{\mathbf{H}_\tau^{*-1}} \right) - \left( \langle x_t, \widehat{\theta}_\tau \rangle - \beta \|x_t\|_{\mathbf{H}_\tau^{*-1}} \right) \\
& \hspace{4cm} \text{(optimistic } x_t \text{, see line 13 algo. 2)} \\
&= 2\beta \|x_t\|_{\mathbf{H}_\tau^{*-1}} \\
&\leq 2\sqrt{2}\beta \|x_t\|_{\mathbf{H}_t^{*-1}} && \left( \text{Lemma B.13}, \frac{\det(\mathbf{H}_\tau^{-1})}{\det(\mathbf{H}_t^{-1})} = \frac{\det(\mathbf{H}_t)}{\det(\mathbf{H}_\tau)} \right) \leq 2)
\end{aligned}
$$

$\square$

**Lemma B.9.** *The arm set $\mathcal{X}_t'$ obtained after eliminating arms from $\mathcal{X}_t$ (line 12 Algorithm 2), under event $\mathcal{E}_w$, satisfies: (a) $x_t^* \in \mathcal{X}_t'$, (b) $\langle x_t^* - x_t, \theta^* \rangle \leq \frac{4}{R}$*

*Proof.* Suppose $x' = \arg\max_{x \in \mathcal{X}_t} LCB_o(x)$. Now, we have, for all $x \in \mathcal{X}_t$,

$$
\begin{aligned}
|\langle x, \widehat{\theta}_o - \theta^* \rangle| &\leq \|x\|_{\mathbf{H}_w^{*-1}} \left\| \widehat{\theta}_o - \theta^* \right\|_{\mathbf{H}_w^*} && \text{(Cauchy-Schwarz)} \\
&\leq \gamma \|x\|_{\mathbf{H}_w^{*-1}} && \text{(by Lemma B.2)} \\
&\leq \gamma\sqrt{\kappa} \|x\|_{\mathbf{V}^{-1}} && (\mathbf{V} \preceq \kappa\mathbf{H}_w^*)
\end{aligned}
$$

Thus, $UCB_o(x_t^*) = \langle x_t^*, \widehat{\theta}_o \rangle + \gamma\sqrt{\kappa} \|x_t^*\|_{\mathbf{V}^{-1}} \geq \langle x_t^*, \theta^* \rangle \geq \langle x', \theta^* \rangle \geq \langle x', \widehat{\theta}_o \rangle - \gamma\sqrt{\kappa} \|x'\|_{\mathbf{V}^{-1}} = LCB_o(x')$ , where the second inequality is due to optimality of $x_t^*$. Hence, $x_t^*$ is not eliminated, implying $x_t^* \in \mathcal{X}_t'$. This completes the proof of (a).

Since $x_t$ is also in $\mathcal{X}_t'$ (by definition),

$$
\begin{aligned}
UCB_o(x_t) = \langle x_t, \widehat{\theta}_o \rangle + \gamma\sqrt{\kappa} \|x_t\|_{\mathbf{V}^{-1}} &\geq \langle x', \widehat{\theta}_o \rangle - \gamma\sqrt{\kappa} \|x'\|_{\mathbf{V}^{-1}} \\
&\geq \langle x_t^*, \widehat{\theta}_o \rangle - \gamma\sqrt{\kappa} \|x_t^*\|_{\mathbf{V}^{-1}} && (x' \text{ has max } LCB_o(\cdot))
\end{aligned}
$$

Again, using the fact that $\langle x_t^*, \widehat{\theta}_o \rangle \geq \langle x_t^*, \theta^* \rangle - \gamma\sqrt{\kappa} \|x_t^*\|_{\mathbf{V}^{-1}}$ and $\langle x_t, \widehat{\theta}_o \rangle \leq \langle x_t, \theta^* \rangle + \gamma\sqrt{\kappa} \|x_t\|_{\mathbf{V}^{-1}}$, we obtain,

$$
\langle x_t, \theta^* \rangle + 2\gamma\sqrt{\kappa} \|x_t\|_{\mathbf{V}^{-1}} \geq \langle x_t^*, \theta^* \rangle - 2\gamma\sqrt{\kappa} \|x_t^*\|_{\mathbf{V}^{-1}}
$$

which gives us, $\quad \langle x_t^* - x_t, \theta^* \rangle \leq 2\gamma\sqrt{\kappa} \|x_t\|_{\mathbf{V}^{-1}} + 2\gamma\sqrt{\kappa} \|x_t^*\|_{\mathbf{V}^{-1}}$

Finally, since Switching Criterion I is not satisfied in this round, $\|x\|_{\mathbf{V}^{-1}} < \frac{1}{R\gamma\sqrt{\kappa}}$ for all $x \in \mathcal{X}_t$. Plugging this above,

$$
\langle x_t^* - x_t, \theta^* \rangle \leq \frac{4}{R}
$$

$\square$

.

## B.4 Proof of Theorem 4.2

In this subsection, we complete the proof of the regret bound of `RS-GLinCB`(Algorithm 2). We first restate Theoreom 4.2 and then prove it. For every round $t \in [T]$, we use $x_t \in \mathcal{X}_t$ to denote the arm played by the algorithm and $x_t^*$ to denote the best available arm in that round.

**Theorem B.10** (Theorem 4.2). *Given $\delta \in (0,1)$, with probability $\geq 1 - \delta$, the regret of `RS-GLinCB` (Algorithm 2) satisfies $R_T = O\big(d\sqrt{\sum_{t\in[T]} \dot{\mu}(\langle x_t^*, \theta^*\rangle)} \log(RT/\delta) + \kappa d^2 R^5 S^2 \log^2(T/\delta)\big)$.*

*Proof.* Firstly, we will assume throughout the proof that $\mathcal{E}_w \cap \mathcal{E}_\tau$ holds, which happens with probability at least $1 - \delta$. Thus, regret of Algorithm 2 is upper bounded as:

$$
\begin{aligned}
R_T &= \sum_{t\in[T]} \mu(\langle x_t^*, \theta^*\rangle) - \mu(\langle x_t, \theta^*\rangle) \\
&\leq R\tau_o + \sum_{t\in[T]\backslash\mathcal{T}_o} \mu(\langle x_t^*, \theta^*\rangle) - \mu(\langle x_t, \theta^*\rangle) \qquad \text{(Upper bound of $R$ for rounds in $\mathcal{T}_o$)} \\
&\leq \mathbf{c}R^3\kappa\gamma^2\log(T/\delta) + \sum_{t\in[T]\backslash\mathcal{T}_o} \dot{\mu}(z)\langle x_t^* - x_t, \theta^*\rangle \\
&\qquad\qquad\qquad\qquad\qquad \text{(some $z \in [\langle x_t, \theta^*\rangle, \langle x_t^*, \theta^*\rangle]$; lemma B.11)}
\end{aligned}
$$

Now, let $R_1(T) = \sum_{t\in[T]\backslash\mathcal{T}_o} \dot{\mu}(z)\langle x_t^* - x_t, \theta^*\rangle$. Hereon, we will slightly abuse notation $\mathbf{H}_\tau$ to denote the $\mathbf{H}_\tau$ matrix last updated before time $t$ for each time step $t \in [T]$. This will be clear from the context as we will only use $\mathbf{H}_\tau$ term-wise. With this, we upper bound $R_1(T)$ as follows:

$$
\begin{aligned}
R_1(T) &\leq \sum_{t\in[T]\backslash\mathcal{T}_o} \dot{\mu}(z)2\beta\|x_t\|_{\mathbf{H}_\tau^{-1}} && \text{(by Lemma B.8)} \\
&\leq \sqrt{2}\beta \sum_{t\in[T]\backslash\mathcal{T}_o} \dot{\mu}(z)2\|x_t\|_{\mathbf{H}_t^{-1}} && \left(\text{Lemma B.13, } \frac{\det(\mathbf{H}_\tau^{-1})}{\det(\mathbf{H}_t^{-1})} = \frac{\det(\mathbf{H}_t)}{\det(\mathbf{H}_\tau)}\right) \leq 2) \\
&\leq 2\sqrt{2}\beta \sum_{t\in[T]\backslash\mathcal{T}_o} \dot{\mu}(z)e^1\|x_t\|_{\mathbf{H}_t^{*-1}} && \text{(by Lemma B.4)} \\
&\leq 2e\sqrt{2}\beta \sum_{t\in[T]\backslash\mathcal{T}_o} \sqrt{\dot{\mu}(\langle x_t^*, \theta^*\rangle)\dot{\mu}(\langle x_t, \theta^*\rangle)} \exp(R\langle x_t^* - x_t, \theta^*\rangle)\|x_t\|_{\mathbf{H}_t^{*-1}} \\
&&& \text{(by Lemma C.2)} \\
&\leq 2e\sqrt{2}\beta \sum_{t\in[T]\backslash\mathcal{T}_o} \sqrt{\dot{\mu}(\langle x_t^*, \theta^*\rangle)}\sqrt{\dot{\mu}(\langle x_t, \theta^*\rangle)}e^4\|x_t\|_{\mathbf{H}_t^{*-1}} && \text{(by Lemma B.9)} \\
&= 2e^5\sqrt{2}\beta \sum_{t\in[T]\backslash\mathcal{T}_o} \sqrt{\dot{\mu}(\langle x_t^*, \theta^*\rangle)}\sqrt{\dot{\mu}(\langle x_t, \theta^*\rangle)}\|x_t\|_{\mathbf{H}_t^{*-1}} \\
&= 2e^5\sqrt{2}\beta \sum_{t\in[T]\backslash\mathcal{T}_o} \sqrt{\dot{\mu}(\langle x_t^*, \theta^*\rangle)}\|\tilde{x}_t\|_{\mathbf{H}_t^{*-1}} && (\tilde{x}_t = \sqrt{\dot{\mu}(\langle x_t, \theta^*\rangle)}x_t) \\
&\leq 2e^5\sqrt{2}\beta \sqrt{\left(\sum_{t\in[T]\backslash\mathcal{T}_o} \dot{\mu}(\langle x_t^*, \theta^*\rangle)\right) \cdot \sum_{t\in[T]\backslash\mathcal{T}_o} \|\tilde{x}_t\|_{\mathbf{H}_t^{*-1}}^2} && \text{(Cauchy-Schwarz)} \\
&\leq 2e^5\sqrt{2}\beta \sqrt{\left(\sum_{t\in[T]\backslash\mathcal{T}_o} \dot{\mu}(\langle x_t^*, \theta^*\rangle)\right) \cdot 2d\log\left(1 + \frac{RT}{\lambda d}\right)} && \text{(Lemma B.12; $\|\tilde{x}_t\|_2 \leq R$)} \\
&\leq \mathbf{c}d\log(RT/\delta)\sqrt{\sum_{t\in[T]\backslash\mathcal{T}_o} \dot{\mu}(\langle x_t^*, \theta^*\rangle)}.
\end{aligned}
$$

Putting things back,

$$\mathrm{R}_T \leq \mathbf{c}d \log(RT/\delta) \sqrt{\sum_{t \in [T] \setminus \mathcal{T}_o} \dot{\mu}\left(\langle x_t^*, \theta^* \rangle\right)} + \mathbf{c}R^5 S^2 \kappa \log(T/\delta)^2.$$

□

## B.5  Bounding number of policy updates: Proof of Lemma 4.1

We first obtain a bound on the number of rounds when Switching Criterion I is satisfied. Then we restate Lemma 4.1 and present its proof. Here, we use $\mathcal{T}_o$ to denote the collection of all rounds till $T$ for which Switching Criterion I is satisfied.

**Lemma B.11.** *Algorithm 2, during its entire execution, satisfies the Switching Criterion I at most* $2dR^2\kappa\gamma^2 \log(T/\delta)$ *times.*

*Proof.* Recall that Switching Criterion I (Line 4) is satisfied, when $\|x\|_{\mathbf{V}^{-1}}^2 > 1/(R^2\kappa\gamma^2)$ for some $x \in \mathcal{X}_t$. Let $\mathbf{V}_m$ be the sequence of $\mathbf{V}$ matrices (line 6 of Algorithm 2) for $m \in \mathcal{T}_o$. That is, $\mathbf{V}_1 = \lambda\mathbf{I}$, $\mathbf{V}_m = \sum_{s \in [m-1] \cap \mathcal{T}_o} x_s x_s^\intercal + \lambda\mathbf{I}$. In these rounds, by Line 5 of Algorithm 2, we have that the arm played $x_t$ is such that $x_t = \arg\max_{x \in \mathcal{X}_t} \|x\|_{\mathbf{V}_t^{-1}}$. Therefore,

$$\sum_{t \in \mathcal{T}_o} \|x_t\|_{\mathbf{V}_t^{-1}}^2 \geq \frac{\tau_o}{R^2\kappa\gamma^2} \tag{22}$$

Furthermore, by the Elliptic Potential Lemma (Lemma B.12) we have

$$\sum_{t \in \mathcal{T}_o} \|x_t\|_{\mathbf{V}_t^{-1}}^2 \leq 2d \log\left(1 + \frac{\tau_o}{\lambda d}\right) \tag{23}$$

Combining (23) and (22) we have

$$\tau_o \leq 2dR^2\kappa\gamma^2 \log\left(1 + \frac{\tau_o}{\lambda d}\right) \leq 2dR^2\kappa\gamma^2 \log(T) \leq 2dR^2\kappa\gamma^2 \log(T/\delta)$$

□

**Lemma** (4.1). *Algorithm 2, during its entire execution, updates its policy at most* $O(R^4 S^2 \kappa d^2 \log^2(T/\delta))$ *times.*

*Proof.* Note that in Algorithm 2, policy changes happen only in the rounds when either of the Switching Criteria are triggered. The number of times Switching Criterion I is triggered is bounded by Lemma B.11. On the other hand, the number of times Switching Criterion II is triggered is equal to the number of times determinant of $\mathbf{H}_t$ doubles, which is bounded by Lemma B.15. Thus in total, the number of policy changes in Algorithm 2 is upper bounded by $2dR^2\kappa\gamma^2 \log(T/\delta) + \mathbf{c}d \log(T)$.  □

## B.6  Some Useful Lemmas

**Lemma B.12** (Elliptic Potential Lemma (Lemma 10 [1])). *Let $x_1, x_2, \ldots x_t$ be a sequence of vectors in $\mathbb{R}^d$ and let $\|x_s\|_2 \leq L$ for all $s \in [t]$. Further, let $\mathbf{V}_s = \sum_{m=1}^{s-1} x_m x_m^\intercal + \lambda\mathbf{I}$. Suppose $\lambda \geq L^2$. Then,*

$$\sum_{s=1}^{t} \|x_s\|_{\mathbf{V}_s^{-1}}^2 \leq 2d \log\left(1 + \frac{L^2 t}{\lambda d}\right) \tag{24}$$

**Lemma B.13** (Lemma 12 of [1]). *Let $A \succeq B \succ 0$. Then*

$$\sup_{x \neq 0} \frac{x^\intercal A x}{x^\intercal B x} \leq \frac{\det(A)}{\det(B)}$$

**Lemma B.14** (Lemma 10 of [1]). *Let $\{x_s\}_{s=1}^t$ be a set of vectors. Define the sequence $\{\mathbf{V}_s\}_{s=1}^t$ as $\mathbf{V}_1 = \lambda\mathbf{I}$, $\mathbf{V}_{s+1} = \mathbf{V}_s + x_s x_s^\intercal$ for $s \in [t-1]$. Further, let $\|x_s\|_2 \leq L \ \forall \ s \in [t]$. Then,*

$$\det(\mathbf{V}_t) \leq \left(\lambda + tL^2/d\right)^d.$$

**Lemma B.15.** *Let $\{x_s\}_{s=1}^t$ be a set of vectors. Define the sequence $\{\mathbf{V}_s\}_{s=1}^t$ as $\mathbf{V}_1 = \lambda\mathbf{I}$, $\mathbf{V}_{s+1} = \mathbf{V}_s + x_s x_s^\mathsf{T}$ for $s \in [t-1]$. Further, let $\|x_s\|_2 \le L \ \forall\ s \in [t]$. Define the set $\{1 = \tau_1, \tau_2 \ldots \tau_m = t\}$ such that: $\det(\mathbf{V}_{\tau_{i+1}}) \ge 2\det(\mathbf{V}_{\tau_i})$ but $\det(\mathbf{V}_{\tau_{i+1}-1}) < 2\det(\mathbf{V}_{\tau_i})$ for $i \in \{2, \ldots m-1\}$. Then, the number of time doubling happens,i.e., $m$, is at most $O(d\log(t))$.*

*Proof.* By Lemma B.14, $\det(\mathbf{V}_t) \le \left(\lambda + tL^2/d\right)^d$. But we have that from definition of $\tau_i$'s

$$
\begin{aligned}
\det(\mathbf{V}_t) &\ge \det(\mathbf{V}_{\tau_{m-1}}) \\
&\ge 2\det(\mathbf{V}_{\tau_{m-2}}) \\
&\ \vdots \\
&\ge 2^{m-2}\det(\mathbf{V}_{\tau_1}) \\
&= 2^{m-2}\det(\mathbf{V}_1) \\
&= 2^{m-2}\lambda^d \qquad\qquad\qquad\qquad (\mathbf{V}_1 = \lambda\mathbf{I})
\end{aligned}
$$

Thus, $2^{m-2}\lambda^d \le \left(\lambda + tL^2/d\right)^d$ which implies that

$$
2^{m-2} \le \left(1 + \frac{tL^2}{\lambda d}\right)^d
$$

Hence, $m \le O(d\log(t))$ . $\qquad\qquad\qquad\qquad\qquad\qquad\qquad\qquad\qquad\qquad\qquad\square$

## C Useful Properties of GLMs

Recall that a Generalized Linear Model is characterized by a canonical exponential family, *i.e.*, the random variable $r$ has density function $p_z(r) = \exp\left(rz - b(z) + c(r)\right)$, with parameter $z$, log-partition function $b(\cdot)$, and a function $c$. Further, $\dot{b}(z) = \mu(z)$ is also called the *link* function.

Hereon, we will assume that the random variable has a bounded non-negative support, *i.e.*, $r \in [0, R]$ almost surely. Now, we state the following key Lemmas on GLMs

**Lemma C.1** (Self-Concordance for GLMs)**.** *For distributions in the exponential family the function $\mu(\cdot)$ satisfies that for all $z \in \mathbb{R}$, $|\ddot{\mu}(z)| \le R\dot{\mu}(z)$.*

*Proof.* Indeed,

$$
\begin{aligned}
|\dddot{b}(z)| &= |\mathbb{E}[(r - \mathbb{E}[r])^3]| &&\text{(Lemma C.3)} \\
&\le \mathbb{E}\left[|(r - \mathbb{E}[r])^3|\right] &&\text{(Jensen's inequality)} \\
&= \mathbb{E}\left[|r - \mathbb{E}[r]| \cdot (r - \mathbb{E}[r])^2\right] \\
&\le \mathbb{E}[R(r - \mathbb{E}[r])^2] &&(r, \mathbb{E}[r] \in [0, R]) \\
&= R\,\mathbb{E}[(r - \mathbb{E}[r])^2] \\
&= R\ddot{b}(z) &&\text{(Lemma C.3)}
\end{aligned}
$$

$\qquad\qquad\qquad\qquad\qquad\qquad\qquad\qquad\qquad\qquad\qquad\qquad\qquad\qquad\qquad\qquad\qquad\square$

As a consequence, we have the following simple modification of the self-concordance results of [6].

**Lemma C.2.** *For an exponential distribution with log-partition function $b(\cdot)$, for all $z_1, z_2 \in \mathbb{R}$, letting $\mu(z) := \dot{b}(z)$, following holds:*

$$
\alpha(z_1, z_2) := \int_{v=0}^1 \dot{\mu}\left(z_1 + v\left(z_2 - z_1\right)\right) \ge \frac{\dot{\mu}(z)}{1 + R|z_1 - z_2|} \quad \text{for } z \in \{z_1, z_2\} \quad (25)
$$

$$
\frac{\dot{\mu}(z_2)}{e^{R|z_2 - z_1|}} \le \dot{\mu}(z_1) \le e^{R|z_2 - z_1|}\dot{\mu}(z_2) \tag{26}
$$

$$
\tilde{\alpha}(z_1, z_2) := \int_{v=0}^1 (1-v)\dot{\mu}\left(z_1 + v(z_2 - z_1)\right) dv \ge \frac{\dot{\mu}(z_1)}{2 + R|z_1 - z_2|} \tag{27}
$$

*Proof.* Without loss of generality, assume that $z_2 \geq z_1$. Note that by property of integration $\int_a^b f(x)dx = \int_b^a f(b+a-x)dx$, $\alpha(z_1, z_2) = \alpha(z_2, z_1)$. Now, by proposition C.1, and the fact that $\ddot{\mu}(z) = \dddot{b}(z)$, we have for any $v \in \mathbb{R}$ and $z \geq z_1$,

$$-R\dot{\mu}(v) \leq \ddot{\mu}(v) \leq R\dot{\mu}(v) \qquad \text{(Lemma C.1)}$$

$$-R \leq \frac{\ddot{\mu}(v)}{\dot{\mu}(v)} \leq R$$

$$-R \int_z^{z_1} dv \leq \int_z^{z_1} \frac{\ddot{\mu}(v)}{\dot{\mu}(v)} dv \leq R \int_z^{z_1} dv$$

$$-R(z - z_1) \leq \log\left(\frac{\dot{\mu}(z)}{\dot{\mu}(z_1)}\right) \leq R(z - z_1)$$

$$\dot{\mu}(z_1)\exp(-R(z - z_1)) \leq \dot{\mu}(z) \leq \dot{\mu}(z_1)\exp(R(z - z_1))$$

Putting $z = z_2$ establishes 26. To show 25, we further set $z = z_1 + u(z_2 - z_1)$ for $u \in [0, 1]$, (note that $z \geq z_1$) and integrate on $u$,

$$\dot{\mu}(z_1) \int_0^1 \exp(-Ru(z_2 - z_1))du \leq \int_0^1 \dot{\mu}(z_1 + u(z_2 - z_1))du \leq \int_0^1 \exp(Ru(z_2 - z_1))du$$

which gives $\quad \dot{\mu}(z_1) \dfrac{1 - \exp(-R(z_2 - z_1))}{R(z_2 - z_1)} \leq \alpha(z_1, z_2) \leq \dot{\mu}(z_1)\dfrac{\exp(R(z_2 - z_1)) - 1}{R(z_2 - z_1)}$

Next, we use the fact that for $x > 0$, $e^{-x} \leq (1 + x)^{-1}$ which on rearranging gives $(1 - e^{-x})/x \geq 1/(1 + x)$. Applying this inequality to the LHS above finishes the proof. Note that similar exercise can be repeated with $z_2 \leq z_1$ to get the same result for $z_2$.

For 27, we have, by application of 26, $\dot{\mu}(z_1 + v(z_2 - z_1)) \geq \dot{\mu}(z_1)\exp(R|v(z_2 - z_1)|)$. Therefore,

$$\begin{aligned}
\tilde{\alpha}(z_1, z_2) &= \int_{v=0}^1 (1 - v)\dot{\mu}(z_1 + v(z_2 - z_1))\, dv \\
&\geq \int_{v=0}^1 (1 - v)\dot{\mu}(z_1)\exp(-R|v(z_1 - z_2)|)dv \\
&= \dot{\mu}(z_1)\int_{v=0}^1 (1 - v)\exp(-Rv|(z_1 - z_2)|)dv \qquad (v \in [0, 1]) \\
&= \dot{\mu}(z_1)\left(\frac{1}{R|z_1 - z_2|} + \frac{\exp(-R|z_1 - z_2|) - 1}{R^2|z_1 - z_2|^2}\right) \\
&\geq \dot{\mu}(z_1) \cdot \frac{1}{2 + R|z_1 - z_2|} \qquad \text{(Lemma 10 of [2])}
\end{aligned}$$

$\square$

Next we state some nice properties of the GLM family that is the key in deriving Lemma C.1.

**Lemma C.3** (Properties of GLMs). *For any random variable $r$ that is distributed by a canonical exponential family, we have*

1. $\mathbb{E}[r] = \mu(z) = \dot{b}(z)$

2. $\mathbb{V}[r] = \mathbb{E}\left[(r - \mathbb{E}[r])^2\right] = \dot{\mu}(z) = \ddot{b}(z)$

3. $\mathbb{E}\left[(r - \mathbb{E}[r])^3\right] = \dddot{b}(z)$

*Proof.*  1. Indeed, since $p_z(r)$ is a probability distribution, $\int_r p_z(r)dr = 1$ which in turn implies that $b(z) = \log\left(\int_r \exp(rz + c(r))dr\right)$. Thus, taking derivative,

$$
\begin{aligned}
\dot{b}(z) &= \frac{1}{\int_r \exp(rz + c(r))dr} \int_r \frac{\partial}{\partial z}\exp(rz + c(r))dr \\
&= \exp(-b(z)) \int_r r\exp(rz + c(r))dr \\
&= \int_r r\exp(rz - b(z) + c(r))dr = \mathbb{E}[r]
\end{aligned}
$$

2. Let $f(z) := \int_r r\exp(rz + c(r))dr$. Thus, $\dot{b}(z) = \exp(-b(z))f(z)$. Taking derivative on both sides,

$$
\begin{aligned}
\ddot{b}(z) &= -\dot{b}(z)\exp(-b(z))f(z) + \exp(-b(z))\dot{f}(z) \\
&= -\mathbb{E}[r]^2 + \exp(-b(z))\int_r r^2\exp(rz + c(r))dr \\
&= -\mathbb{E}[r]^2 + \int_r r^2\exp(rz - b(z) + c(r))dr \\
&= -\mathbb{E}[r]^2 + \mathbb{E}[r^2] = \mathbb{V}[r]
\end{aligned}
$$

3. Again let $f(z) := \int_r r^2\exp(rz + c(r))dr$. Thus, $\ddot{b}(z) = -\dot{b}(z)^2 + \exp(-b(z))f(z)$. Taking derivative on both sides,

$$
\begin{aligned}
\dddot{b}(z) &= -2\dot{b}(z)\ddot{b}(z) - \dot{b}(z)\exp(-b(z))f(z) + \exp(-b(z))\dot{f}(z) \\
&= -2\dot{b}(z)\ddot{b}(z) - \dot{b}(z)\mathbb{E}[r^2] + \int_r r^3\exp(rz - b(z) + c(r))dr \\
&= -2\mathbb{E}[r]\mathbb{V}[r] - \mathbb{E}[r]\mathbb{E}[r^2] + \mathbb{E}[r^3]
\end{aligned}
$$

Now, let us expand $\mathbb{E}[(r - \mathbb{E}[r])^3]$.

$$
\begin{aligned}
\mathbb{E}[(r - \mathbb{E}[r])^3] &= \mathbb{E}[r^3 - 3r^2\mathbb{E}[r] + 3r\mathbb{E}[r]^2 - \mathbb{E}[r]^3] \\
&= \mathbb{E}[r^3] - 3\mathbb{E}[r]\mathbb{E}[r^2] + 3\mathbb{E}[r]^3 - \mathbb{E}[r]^3 \\
&= \mathbb{E}[r^3] - \mathbb{E}[r]\mathbb{E}[r^2] - 2\mathbb{E}[r]\left(-\mathbb{E}[r^2] + \mathbb{E}[r]^2\right) \\
&= \mathbb{E}[r^3] - \mathbb{E}[r]\mathbb{E}[r^2] - 2\mathbb{E}[r]\mathbb{V}[r]
\end{aligned}
$$

$\square$

**Corollary C.4.** *For all exponential family, $b(\cdot)$ is a convex function.*

*Proof.* Indeed, note that $\ddot{b}(z) = \mathbb{V}[r]$ which is always non-negative. Thus, $\ddot{b}(z) \geq 0$ implying that $b(\cdot)$ is convex. $\square$

*Remark C.5.* In [5] Section 1.4.1, the author claims that if the GLM parameter $z$ lies in a bounded set, then the GLM is self-concordant, i.e., $|\ddot{\mu}(z)| \leq a\dot{\mu}(z)$, for some appropriate constant $a$ over this bounded set. Thereafter the author notes that the techniques developed in [5] guarantees $\kappa$-free regret rates (in $\sqrt{T}$ term) for such GLMs (i.e., all GLMs with bounded parameter). However, the claim regarding self-concordance of GLMs is not true in general. There are classes of GLMs whose parameters may be restricted in a bounded set, but for them no constant $a$ exists. One such example is the exponential distribution. The link function $\mu$ for exponential distribution is given as $\mu(z) = -\frac{1}{z}$. If we allow $z$ to lie in the set $(-c, 0)$ for some positive $c$, then we have $\mu(z)$ strictly increasing (satisfying our assumption on monotonicity of $\mu$, thus a valid example). However, for this GLM,

$$
\dot{\mu}(z) = \frac{1}{z^2} \qquad \ddot{\mu}(z) = -\frac{2}{z^3}
$$

Note that $\ddot{\mu}(z)$ is positive for the assumed support of $z$. Suppose this GLM is self-concordant, then we must have some positive constant $a$ such that

$$|\ddot{\mu}(z)| = -\frac{2}{z^3} \leq a\dot{\mu}(z) = a\frac{1}{z^2} .$$

Simplifying, we obtain the following relation:

$$-\frac{2}{z} \leq a .$$

However, since $z \in (-c, 0)$, we have $\lim_{z \to 0} -\frac{2}{z} \to \infty$. Hence, no constant $a$ is possible. By this counterexample it can be seen that bounded parameter set is not enough to guarantee self-concordance of GLMs. In this work, we give a characterization of self-concordance of GLMs with bounded support of the random variable. It will be interesting to understand a complete characterization of self-concordance of GLMs.

## D  Computational Cost

Consider a log-loss minimization oracle that returns the unconstrained MLE for a given GLM class with a computational complexity of $C_{opt} \cdot n$, when the log-loss is computed over $n$ data points. Let the maximum number of arms available every round be $K$. Furher, let the computational cost of an oracle that solves the non-convex optimization 8 be $NC_{opt}$.

**Computational Cost of** B-GLinCB: In the B-GLinCB algorithm, we employ the log-loss oracle at the end of each batch. The estimator $\widehat{\theta}$ calculated at the end of a batch of length $\tau$ incurs a computational cost of $C_{opt}\tau$. Furthermore, this oracle is invoked for a maximum of $M \leq \log\log T$ batches. Additionally, the computation of the distributional optimal design at the end of each batch is efficient in $d$ (poly$(d)$). Moreover, in every round, the algorithm solves the $D/G-$ Optimal Design problem (requiring $O(d\log d)$ computation) and runs elimination based on prior (at most $\log\log T$) phases. Hence, the amortized cost per round of B-GLinCB is $O(K\log\log T + d\log d + C_{opt})$.

**Computational Cost of** RS-GLinCB: In the RS-GLinCB algorithm, the estimator $\widehat{\theta}_o$ is computed each time Switching Criterion I is triggered. Additionally, during rounds when Switching Criterion I is not triggered, the estimator $\widehat{\theta}_\tau$ is computed a maximum of $O(\log(T))$ times. These computations involve utilizing both the log-loss oracle and the non-convex projection oracle. Furthermore, in each non-Switching Criterion I round, the algorithm executes an elimination step. This yields an amortized time complexity of $O(C_{opt}\log(T) + NC_{opt}\log^2(T) + K)$ per round.

**Performance in Practice**: As evident from Fig. 1, RS-GLinCB has much better computational performance in practice. We ran all the experiments on an Azure Data Science VM equipped with AMD EPYC 7V13 64-Core Processor (clock speed of 2.45 GHz) and Linux Ubuntu 20.04 LTS operating system. It was ensured that no other application processes were running while we tested the performance. We implemented and tested our code in Python, and measured the execution times using `time.time()` command. We allowed no operations for 10 seconds after every run to let the CPU temperature come back to normal, in case the execution heats up the CPU, thereby causing subsequent runs to slow down.

Comparison with ECOLog [7] shows that execution time for RS-GLinCB is significantly smaller. We posit that this is because RS-GLinCB solves a large convex optimization problem but less frequently, resulting into smaller overhead at the implementation level, while ECOLog solves a smaller convex optimization problem, but does so every round. On an implementation level, this translates into more function calls and computation. Further, we observe that with increasing $\kappa$, the execution time of RS-GLinCB increases, which is in accordance with Lemma 4.1 that quantifies the number of policy switches as an increasing function of $\kappa$.

While comparing with GLOC [14], we observe that RS-GLinCB performs better than GLOC in both high and low $\kappa$ regimes. Since GLOC runs an online convex optimization (online Newton step) algorithm to generate its confidence sets, the time taken by GLOC is nearly constant with changing $\kappa$. On the other hand, in accordance with Lemma 4.1, the computational cost of RS-GLinCB increases with $\kappa$. However, after a few initial rounds, when neither of the switching criteria are triggered, RS-GLinCB does not need to solve any computationally intensive optimization problem, hence these rounds execute very fast. In practice, with typical data distribution, RS-GLinCB reaches this stage much before what the worst-case guarantees show, hence we see it perform better than GLOC.

# E Projection

We describe the projection step used in Algorithms 1 and 2. We present arguments similar to the ones made in Appendix B.3 of [6]. We write

$$\mathbf{H}(\theta) = \sum_{s=1}^{t} \dot{\mu}\left(\langle \theta, x_s \rangle\right) x_s x_s^\mathsf{T} + \lambda \mathbf{I}$$

Recall, $\mathbf{H}^* = \mathbf{H}(\theta^*)$. Let $\widehat{\theta}$ be the MLE estimator of $\theta^*$ calculated after the sequence arm pulls $x_1, x_2, \ldots, x_t$. Let $r_1, r_2, \ldots, r_t$ be the corresponding observed rewards. We project $\widehat{\theta}$ to a set $\Theta$ by solving the following optimization problem

$$\widetilde{\theta} := \arg\min_{\theta \in \Theta} \left\| \sum_{s=1}^{t} (\mu\left(\langle x_s, \theta \rangle\right) - \mu\left(\langle x_s, \widehat{\theta} \rangle\right)) x_s \right\|_{\mathbf{H}(\theta)^{-1}} \tag{28}$$

**Lemma E.1.** *Using the notations described above, if $\theta^* \in \Theta$ and $\max_{i \in [t]} |\langle x_i, \widetilde{\theta} - \theta^* \rangle| \leq c/R$, then we have*

$$\left\| \widetilde{\theta} - \theta^* \right\|_{\mathbf{H}(\theta^*)} \leq 2(1+c) \left\| \sum_{s=1}^{t} (\mu\left(\langle x_s, \theta^* \rangle\right) - r_s) x_s \right\|_{\mathbf{H}(\theta^*)^{-1}}$$

*Proof.* First, we note that by self-concordance property of $\mu$ (lemma C.2), for any $s \in [t]$,

$$\begin{aligned}
\alpha(x_s, \widetilde{\theta}, \theta^*) &\geq \frac{\dot{\mu}\left(\langle x_s, \theta^* \rangle\right)}{1 + R|\langle x_s, \widetilde{\theta} - \theta^* \rangle|} \\
&\geq \frac{\dot{\mu}\left(\langle x_s, \theta^* \rangle\right)}{1 + R(c/R)} \qquad (\max_{i \in [s]} |\langle x_s, \widetilde{\theta} - \theta^* \rangle| \leq c/R) \\
&= \frac{\dot{\mu}\left(\langle x_s, \theta^* \rangle\right)}{1 + c}
\end{aligned}$$

Similarly, we have $\alpha(x_s, \widetilde{\theta}, \theta^*) \geq \frac{\dot{\mu}\left(\langle x_s, \widetilde{\theta} \rangle\right)}{1+c}$.

Let us define the matrix $\mathbf{G} = \sum_{s \in [t]} \alpha(x, \widetilde{\theta}, \theta^*) x_s x_s^\mathsf{T}$. Using the above fact, we obtain the relation: $\mathbf{G} \succeq \frac{1}{1+c} \mathbf{H}^*$ and $\mathbf{G} \succeq \frac{1}{1+c} \mathbf{H}(\widetilde{\theta})$. Also define the vector $g(\theta) = \sum_{s \in [t]} \mu\left(\langle \theta, x_s \rangle\right) x_s$. Now,

$$\begin{aligned}
\left\| \widetilde{\theta} - \theta^* \right\|_{\mathbf{H}^*} &\leq \sqrt{1+c} \left\| \widetilde{\theta} - \theta^* \right\|_{\mathbf{G}} &&(\mathbf{H}^* \preceq (\sqrt{1+c})\mathbf{G}) \\
&= \sqrt{1+c} \left\| \mathbf{G}\left(\widetilde{\theta} - \theta^*\right) \right\|_{\mathbf{G}^{-1}} \\
&= \sqrt{1+c} \left\| \sum_{s \in [t]} \left( \alpha(x_s, \widetilde{\theta}, \theta^*) \langle \widetilde{\theta} - \theta^*, x_s \rangle \right) x_s \right\|_{\mathbf{G}^{-1}} \\
&= \sqrt{1+c} \left\| \sum_{s \in [t]} \left( \mu\left(\langle x_s, \widetilde{\theta} \rangle\right) - \mu\left(\langle x_s, \theta^* \rangle\right) \right) x_s \right\|_{\mathbf{G}^{-1}} &&\text{(Taylor's theorem)} \\
&= \sqrt{1+c} \left\| \left( \sum_{s \in [t]} \mu\left(\langle \widetilde{\theta}, x_s \rangle\right) x_s \right) - \left( \sum_{s \in [t]} \mu\left(\langle \theta^*, x_s \rangle\right) x_s \right) \right\|_{\mathbf{G}^{-1}}
\end{aligned}$$

Let $g(\theta) = \sum_{s=1}^{t} \dot{\mu}\left(\langle x_s, \theta \rangle\right) x_s$ for any $\theta$. Therefore, we have,

$$
\begin{aligned}
\left\|\widetilde{\theta} - \theta^*\right\|_{\mathbf{H}^*} &\leq \sqrt{1+c}\left\|g(\widetilde{\theta}) - g(\theta^*)\right\|_{\mathbf{G}^{-1}} \\
&= \sqrt{1+c}\left\|g(\widetilde{\theta}) - g(\widehat{\theta}) + g(\widehat{\theta}) - g(\theta^*)\right\|_{\mathbf{G}^{-1}} \\
&\leq \sqrt{1+c}\left(\left\|g(\widetilde{\theta}) - g(\widehat{\theta})\right\|_{\mathbf{G}^{-1}} + \left\|g(\widehat{\theta}) - g(\theta^*)\right\|_{\mathbf{G}^{-1}}\right) \qquad (\triangle \text{ inequality}) \\
&\leq (1+c)\left(\left\|g(\widetilde{\theta}) - g(\widehat{\theta})\right\|_{\mathbf{H}(\widetilde{\theta})^{-1}} + \left\|g(\widehat{\theta}) - g(\theta^*)\right\|_{\mathbf{H}^{*-1}}\right) \\
&\qquad\qquad\qquad\qquad\qquad\qquad\qquad (\mathbf{H}^{*-1} \succeq (\sqrt{1+c})\mathbf{G}^{-1}) \\
&\leq 2(1+c)\left\|g(\widehat{\theta}) - g(\theta^*)\right\|_{\mathbf{H}^{*-1}} \qquad\qquad\qquad\qquad (\text{by } (28)) \\
&= 2(1+c)\left\|g(\theta^*) - \sum_{s\in[t]} r_s x_s\right\|_{\mathbf{H}^{*-1}} \\
&\qquad\qquad\qquad (\widehat{\theta} \text{ is the unconstrained MLE, } g(\widehat{\theta}) = \sum_{s\in[t]} r_s x_s.)
\end{aligned}
$$

$\square$

## E.1 Convex Relaxation

The optimization problem in (28) is a non-convex optimization problem and therefore it is not clear what is the computational complexity of the problem. However, it is possible to substitute this optimization problem with a convex one, whose computational complexity can be better tractable. The process is similar to the one detailed in [2, section 6]. Here we briefly outline the steps.

Let $\mathcal{L}_t(\theta) = \sum_{s=1}^{t} \ell(\theta, x_s, r_s)$ and $\breve{\theta}$ be defined as follows:

$$
\breve{\theta} := \arg\min_{\theta \in \Theta} \mathcal{L}_t(\theta) \tag{29}
$$

Note that when the set $\Theta$ is a convex set, then the above optimization problem is convex by property of the log-likelihood function of GLMs. Hence it can be solved efficiently. With this projected $\breve{\theta}$, we have the following guarantee:

**Lemma E.2.** *Suppose* $\left\|g(\widehat{\theta}) - g(\theta^*)\right\|_{\mathbf{H}^{*-1}} \leq \gamma$ *and* $\lambda = \gamma/R$. *If* $\theta^* \in \Theta$ *and* $\max_{i\in[t]} |\langle x_i, \breve{\theta} - \theta^* \rangle| \leq c/R$, *then we have*

$$
\left\|\breve{\theta} - \theta^*\right\|_{\mathbf{H}(\theta^*)} \leq c\sqrt{(2+c)R^3 S\gamma}
$$

*Proof.* First we note that by self-concordance property of $\mu$, for any $s \in [t]$,

$$
\begin{aligned}
\tilde{\alpha}(x_s, \theta^*, \breve{\theta}) &\geq \frac{\dot{\mu}\left(\langle x_s, \theta^* \rangle\right)}{2 + R|\langle x_s, \breve{\theta} - \theta^* \rangle|} \qquad\qquad (\text{Lemma C.2}) \\
&\geq \frac{\dot{\mu}\left(\langle x_s, \theta^* \rangle\right)}{2 + R(c/R)} \qquad\qquad (\max_{i\in[s]} |\langle x_s, \widetilde{\theta} - \theta^* \rangle| \leq c/R) \\
&= \frac{\dot{\mu}\left(\langle x_s, \theta^* \rangle\right)}{2 + c}
\end{aligned}
$$

Let us define $\tilde{\mathbf{G}}(\theta^*, \theta) := \sum_{s=1}^{t} \tilde{\alpha}(x_s, \theta^*, \theta) x_s x_s^\mathsf{T}$. Using the above fact, we obtain $\tilde{\mathbf{G}}(\theta^*, \theta) \succeq \frac{1}{2+c}\mathbf{H}^*$.

We now follow closely the proof outlined in Appendix B.3 of [2] with minor changes. By second-order Taylor's expansion, for any $\theta \in \mathbb{R}^d$, we can write

$$\mathcal{L}_t(\theta) - \mathcal{L}_t(\theta^*) - \langle \nabla \mathcal{L}_t(\theta^*), \theta - \theta^* \rangle = \|\theta - \theta^*\|^2_{\check{\mathbf{G}}(\theta, \theta^*)}$$

$$\geq \frac{1}{2+c} \|\theta - \theta^*\|^2_{\mathbf{H}^*}$$

Taking absolute value on both sides, and substituting $\theta = \breve{\theta}$,

$$\left\|\breve{\theta} - \theta^*\right\|^2_{\mathbf{H}^*} \leq (2+c)\left(|\mathcal{L}_t(\breve{\theta}) - \mathcal{L}_t(\theta^*)| + |\langle \nabla \mathcal{L}_t(\theta^*), \breve{\theta} - \theta^* \rangle|\right) \qquad (\triangle\text{-inequality})$$

$$\leq (2+c)\left(|\mathcal{L}_t(\breve{\theta}) - \mathcal{L}_t(\theta^*)| + \|\nabla \mathcal{L}_t(\theta^*)\|_{\mathbf{H}^{*-1}}\left\|\breve{\theta} - \theta^*\right\|_{\mathbf{H}^*}\right) \quad \text{(Cauchy-Schwarz)}$$

$$= (2+c)\left(|\mathcal{L}_t(\breve{\theta}) - \mathcal{L}_t(\theta^*)| + \left\|g(\theta^*) - \sum_{s \in [t]} r_s x_s\right\|_{\mathbf{H}^{*-1}}\left\|\breve{\theta} - \theta^*\right\|_{\mathbf{H}^*}\right)$$

Recall that $\widehat{\theta}$ is the unconstrained MLE, therefore $\nabla \mathcal{L}_t(\widehat{\theta}) = \mathbf{0}$. By a similar Taylor expansion as above and some algebraic manipulations (see Appendix B.3 of [2]), we have, for $\theta^*$,

$$\mathcal{L}_t(\theta^*) - \mathcal{L}_t(\widehat{\theta}) \leq \left\|g(\theta^*) - g(\widehat{\theta})\right\|^2_{\mathbf{G}(\theta^*, \widehat{\theta})^{-1}}$$

$$\leq \frac{R}{\sqrt{\lambda}}\left\|g(\theta^*) - g(\widehat{\theta})\right\|^2_{\mathbf{H}^{*-1}} + \left\|g(\theta^*) - g(\widehat{\theta})\right\|_{\mathbf{H}^{*-1}}$$

$$\leq \frac{R}{\sqrt{\lambda}}\gamma^2 + \gamma \qquad\qquad\qquad\qquad\qquad\qquad \text{(Lemma B.1)}$$

$$\leq 2R^3 S\gamma \qquad\qquad\qquad\qquad\qquad\qquad (\text{recall } \sqrt{R^2\lambda} = \gamma/RS)$$

We also have, by definition of $\breve{\theta}$, whenever $\theta^* \in \Theta$, $\mathcal{L}_t(\breve{\theta}) \leq \mathcal{L}_t(\theta^*)$, therefore we have $\mathcal{L}_t(\breve{\theta}) - \mathcal{L}_t(\widehat{\theta}) \leq \mathcal{L}_t(\theta^*) - \mathcal{L}_t(\widehat{\theta}) \leq 2R^3 S\gamma$ Thus, we have,

$$\left\|\breve{\theta} - \theta^*\right\|^2_{\mathbf{H}^*} \leq (2+c)\left(4R^3 S\gamma + \gamma\left\|\breve{\theta} - \theta^*\right\|_{\mathbf{H}^*}\right)$$

Using the inequality that for some $x^2 \leq bx + c \implies x \leq b + \sqrt{c}$, we have,

$$\left\|\breve{\theta} - \theta^*\right\|_{\mathbf{H}^*} \leq (2+c)\gamma + \sqrt{(2+c)4R^3 S\gamma}$$

$$= \mathbf{c}\sqrt{(2+c)R^3 S\gamma}$$

$\square$

