# OpenReview forum: "Generalized Linear Bandits with Limited Adaptivity"
_NeurIPS.cc/2024/Conference — NeurIPS 2024 spotlight_

### Official Review · Reviewer_5bc4 · 2024-07-03

**Soundness:** 3
**Presentation:** 3
**Contribution:** 3
**Rating:** 5
**Confidence:** 2

**Summary:**

This paper addresses the generalized linear contextual bandit problem under limited adaptivity constraints. In a setting (M1) where the times for updating the agent's policy are predetermined, the first proposed algorithm B-GLinCB divides the entire timeline into batches, updating the policy at the end of each batch. B-GLinCB explores using a G-optimal design policy in the first batch and adjusts the arm set in subsequent batches based on estimated MLE parameters using samples obtained in the first batch. The algorithm guarantees a $\tilde{O}(\sqrt{T})$ regret when the number of policy updates is $\Omega(\log \log T)$, independent of the problem-dependent instance $\kappa$. In a setting (M2) where the agent can adaptively decide when to update the policy, the RS-GLinCB algorithm is proposed. RS-GLinCB uses two criteria to alter action selection: the first criterion allows a tighter estimation of the true parameter's derivative, and the second criterion achieves a $\kappa$-independent regret bound. The theoretical results of the proposed algorithms were supported through comparisons with baseline algorithms in logistic bandit settings.

**Strengths:**

- The limited adaptivity constraint discussed in the proposed paper is crucial for many real-world decision-making problems. The authors extend results from linear reward models to non-linear reward models, with the proposed algorithms guaranteeing $\tilde{O}(\sqrt{T})$ regret under specific conditions.
- Notably, the leading term of the regret bound for the proposed algorithms is independent of problem-dependent instances. This is, to my knowledge, the first result showing kappa-independent regret bounds for GLM reward models, excluding logistic bandits.
- The proposed algorithms are computationally efficient since the number of samples used to estimate the reward parameter does not increase over time.

**Weaknesses:**

- Although the proposed algorithm achieves a $\kappa$-independent regret bound, it requires prior knowledge of $\kappa$. The MNL contextual bandit (Perivier & Goyal, 2022) achieved a $\kappa$-independent regret bound without needing information about $\kappa$, which might be useful here.
- The proposed algorithm is computationally efficient concerning time $t$, but there is no explanation of its dependency on the dimension $d$. It lacks details on the computational complexity needed for calculating the optimal design and distributional optimal design at each time step.

* * *
Perivier & Goyal. "Dynamic pricing and assortment under a contextual MNL demand." Advances in Neural Information Processing Systems 35 (2022): 3461-3474.

**Questions:**

1. In logistic bandits[2, 6, 7], algorithms achieve $\kappa$-independent regret bounds by leveraging the self-concordance property of log-loss. What specific challenges were encountered when extending this to GLMs?

2. In the experiments, it appears that ECOLog in [7] is a sub-algorithm for reward parameter estimation. If the comparison is with OFU-ECOLog, which also has a computational complexity of $O(\log t)$, what might explain the significant difference in execution times between OFU-ECOLog and RS-GLinCB?
3. Additionally, how does the execution time difference vary with increasing context dimensions?
4. Why were different values for the upper bound of the reward parameter S used in experiments (S=3 for logistic and S=5 for probit)?

- [Minor typos]
    - The $\kappa$ in line 137 and the $\kappa$ in line 144 seem to refer to different concepts; should different symbols be used?
    - Line 156: G-optimal design policy $\pi_G = \arg \min_\lambda \min_x \| x \|^2_{U(\lambda)^{-1}}$
    - Algorithm 2, line 14: $\hat{\theta}_w\) → \(\hat{\theta}_o$
    - Constraint in Eq. (6): $|| \theta - \hat{\theta}_w ||_V \le \gamma \sqrt{\kappa}$ → $|| \theta - \hat{\theta}_o ||_V \le \gamma \sqrt{\kappa}$

**Limitations:**

The authors have well-addressed the limitations and further research directions in Section 6.

The content discussed in this paper appears to have little to no negative societal impact.

---

> ### Author Rebuttal · Authors · 2024-08-07
>
> We thank the reviewer for the feedback. We address the comments and questions below:
>
>
> **Regarding Weakness 1**: *Prior knowledge of $\kappa$*.
> We note that [7], in fact, assumes the knowledge of an upper bound on $\kappa$ in Procedure 1 for the non-contextual problem and while calculating $\textbf{V}^\mathcal{H}_s$ matrix for the contextual setting. We work under the constraint of limited adaptivity, and our algorithms use the value of $\kappa$ during the warmup round of B-GLinCB and for the first switching criteria in RS-GLinCB. We believe a $\kappa$-dependent warmup round is especially necessary for the batch algorithm.
> As mentioned in response (1a) for Review wYhM, access to an upper bound on $\kappa$ suffices. In particular, as is standard in bandit literature (specifically, linear and generalized linear bandits), one can assume access to an upper bound on $\theta*$. Such an upper bound directly translates into the required upper bound for $\kappa$.
>
> We thank the reviewer for pointing us to the relevant work of Periviar & Goyal, which we will include in the final version of the paper.
>
>
> **Regarding Weakness 2**: *Dependence on context dimension $d$ in computational complexity*.
> A detailed discussion on computational complexity is provided in Appendix D (lines 675-692).
> Our results hold for GLMs, in general. Here, for any class of reward functions (e.g., logistic rewards) for which the specified convex optimization problem can be solved efficiently, we obtain a polynomial dependence on $d$.
>
>
>
> **Regarding Question 1**.
> Our novel instantiation with respect to self concordance is detailed in Remark 1.3 (Lines 93-98).
> Our overarching contribution is the development of GLM bandit algorithms with a key focus on limited adaptivity. This required new algorithmic techniques (e.g., the $\kappa$ dependent exploration phase). To complement the algorithms’ design, the analysis required new ways to adapt the self-concordance property of the reward distribution (e.g. Lemma A.5, A.7). Further,  new technical ideas were required both on the technical (e.g., Lemmas A.16 A.17) and analytic fronts.
>
> **Regarding Question 2**: *"In the experiments, it appears that ECOLog in [7] is a sub-algorithm for reward parameter estimation"*.
> We have discussed the superior empirical performance of RS-GLinCB in detail in Appendix D (lines 693-715).
> There seems to be a factual oversight here: we do not use ECOLog as a sub-algorithm. In fact, ECOLog and our algorithms solve notably different optimization problems for estimating $\theta^*$.
>
> In particular, ECOLog [7] estimates $\theta^*$ by optimizing a second order approximation of the true log-loss while we optimize the true log-loss; however, we do it less frequently.  That is, RG-GLinCB solves a ‘larger’ convex optimization problem but less frequently. This results in a smaller overhead at the implementation level. By contrast, ECOLog solves a smaller optimization problem but does so every round. Moreover, ECOLog solves additional optimization problems every round for the adaptive warmup criteria (for estimating parameter ${\theta}_t^0$, ${\theta}_t^1$ and $\bar{\theta}_t$ in Algorithm 2). We, on the other hand, have a simpler warmup criteria (see Switching Criteria 1) that only relies on the arm geometry and does not require solving an optimization problem.
>
> **Regarding Question 3**.
> Space permitting, we will extend the experiments to highlight the dependence of the execution times on $d$.
>
> **Regarding Question 4**: *Different values of $S$ for logistic and probit rewards*.
> We observe that the empirical performance of RS-GLinCB is consistently better in terms of both regret and computational performance compared to the previous best logistic and other GLM bandit algorithms. The choice of exact parameter value $S, \kappa$ etc., is arbitrary. We will include a comparison with different values of $S$ in the updated version.
>
>
> We thank the reviewer for pointing out the minor typos and will fix them.

---

> > ### Comment · Reviewer_5bc4 · 2024-08-11
> >
> > Thank you for the detailed response. I have no further questions.

---

> > > ### Author Response · Authors · 2024-08-11
> > >
> > > Thank you for acknowledging the response. Do let us know if we can provide additional details which might support increasing your score.

---

### Official Review · Reviewer_bRFJ · 2024-07-09

**Soundness:** 4
**Presentation:** 4
**Contribution:** 4
**Rating:** 8
**Confidence:** 4

**Summary:**

The authors consider the problem of regret minimization in bounded generalized linear contextual bandits with limited adaptivity. Specifically, they consider two models of limited adaptivity: **M1** in which the update rounds must be chosen before the algorithm is run, and **M2** in which the algorithm can be adaptive as the algorithm proceeds. For **M1** the authors propose B-GLinCB that obtains $\tilde{O}(dRS\sqrt{T/\kappa^*})$, and for **M2**, RS-GLinCB that obtains $\tilde{O}(d \sqrt{T/\kappa^*})$. The efficacy of RS-GLinCB is shown numerically.

**Strengths:**

- Clearly and well-written
- First $d \sqrt{T/\kappa}$-type regret that holds for generalized linear bandits *beyond* logistic bandits + $\mathrm{poly}(S)$-free leading term for generalized linear bandits
- Numerous interesting (and important) technical contributions were made to the algorithm design and the regret analysis.

**Weaknesses:**

- The numerical experiments could benefit from more comparators, namely, randomized algorithms: Thompson sampling [1] (and its follow-up works, e.g., [2]) and recently proposed EVILL [3]. I know (and appreciate) that the authors' algorithms are primarily for the limited adaptivity scenarios. Still, given how one of the main contributions of this paper is state-of-the-art regret analysis, it would be important to have these (practically performing well) randomized algorithms as comparators.
- (Continuing from the first point) In the prior regret analyses of logistic bandits by [4], they obtained $\tilde{O}(d\sqrt{T/\kappa^*} + \kappa \wedge R_-(T))$, where $R_-(T)$ is some arm-geometry-adaptive term that can be much smaller than $\kappa$. As the algorithm here makes use of warmup, it must incur the worst-case geometry-dependent term. The authors should also compare with this algorithm for the sake of regret comparison.
- The algorithms involve an explicit warmup stage, which in practice may be not so good. This is shown in the logistic bandits experiments, where although RS-GLinUCB is good eventually, its warmup (that scales with $\kappa$) forces the algorithm to incur high regret in the beginning (til round ~10000).
- Some references on logistic bandits missing, namely [5] where in fixed arm-set setting, they proposed a Hessian-based optimal design (H-optimality), then a warmup-based algorithm was proposed that obtains $\tilde{O}(d \sqrt{T/\kappa^*})$ regret, which is $\mathrm{poly}(S)$-free.
- Although the regret of RS-GLinCB is indeed $\mathrm{poly}(S)$-free, it mainly relies on a nonconvex optimization (Eqn. (6)), and it seems that the tractable convex relaxation again introduces factors of $S$ (and $R$) to the leading term. (please correct me here if I'm wrong) -- This point should be made precise in the introduction.



[1] https://proceedings.mlr.press/v108/kveton20a.html

[2] https://arxiv.org/abs/2209.06983

[3] https://proceedings.mlr.press/v238/janz24a.html

[4] https://proceedings.mlr.press/v130/abeille21a.html

[5] https://arxiv.org/abs/2202.02407

**Questions:**

- The authors mention that the optimization problem in Eqn. (6) is nonconvex, and a convex relaxation results in additional factors of $\mathrm{poly(R, S)}$. Do the factors appear in the leading term as well?
- Are the analyses and algorithms amenable to unbounded, generalized linear models that are self-concordant? For instance, Gaussian is self-concordant with a multiplicative factor of $1$ but is unbounded.
- For the algorithms, the authors use l2-penalized MLE, then project it to the S-ball if necessary. Why not just do l2-constrained MLE, as done in [6]?
- The authors used an optimal design based on $\lVert \cdot \rVert_{V_t}$, thus incurring explicit dependency on $\kappa$. Is there any way to make the warmup more efficient by considering geometry-dependent norm, e.g., $\lVert \cdot \rVert_{H_t}$ as in [5]?
- The authors stated that the ellipsoidal projection is the main technical novelty for obtaining $\mathrm{poly}(S)$-free regret for RS-GLinCB. Can this then be combined with prior UCB-based algorithms for logistic bandits (or GLM bandits) to obtain similar improvements? Or is it the case that such ellipsoidal projection *combined* with some other techniques for the limited adaptivity allows for $\mathrm{poly}(S)$-free regret?
- (minor) Can the intuitions and ideas from kappa-dependent warmup be used for best arm identification, e.g., [7]?



[6] https://proceedings.mlr.press/v238/lee24c.html

[7] https://proceedings.mlr.press/v139/jun21a.html

(If all my concerns are sufficiently addressed, I'm leaning towards further raising the score)

**Limitations:**

Yes

---

> ### Author Rebuttal · Authors · 2024-08-07
>
> We thank the reviewer for the detailed and insightful review.
>
> **Regarding Weakness 1**: *Numerical comparisons with DDRTS-GLB and EVILL*.
> This is a useful suggestion. We will implement the additional empirical comparisons mentioned in the review.
> On the theoretical front, it is, however, relevant to note that DDRTS-GLB is an O(T^2) computation algorithm with non-optimal regret ($\kappa$-dependence). EVILL is not comparable because that work deals with fixed arm sets, not contextual arms.
>
> **Regarding Weakness 2**: *Regarding arm-geometry-adaptive term*.
> Our analysis is not tuned to the arm-geometry dependent $R_-(X)$ term as found in [2] and [17]. This is primarily because our algorithms require a warm-up (implicitly in RS-GLinUCB and explicitly in B-GLinUCB), during which there is no control over the regret, even in the analysis. This conforms with existing literature, for example, [7], where the second-order term is not able to accommodate the arm-geometry dependent $R_-(X)$ term. As future work, it will be interesting to investigate algorithms that are efficient while being able to accommodate arm-geometry dependent regret for GL Bandits.
> We appreciate this point and will highlight it in the updated version.
>
> **Regarding Weakness 3**: *Warm-up in RS-GLinCB*.
> The warm-up in RS-GLinCB is implicit and arm-geometry adaptive. The rounds in which warm-up occurs (i.e., when Switching Criterion I is triggered) are deterministic, based on the sequence of contexts  $\\{ X_t \\}_{t \geq 1}$. This leads to warm-up only in certain rounds based on the geometry of the contexts, which is very different from warm-up in [7], where warm-up is decided based on the stochasticity of rewards as well. Therefore, in our experiments, we observe that after a few initial rounds, the regret is nearly constant, while other algorithms display a growing nature. For the logistic case, among the efficient algorithms available, RS-GLinUCB’s superiority is clearly established.
>
> **Regarding Weakness 4**: *Prior works*.
> Thank you for pointing out the relevant work of Mason et al. 2018. We will include it in our final version.
>
> **Regarding Weakness 5 and Question 1**: *Convex relaxation and dependence on S*.
> The reviewer is right that a convex relaxation leads to poly(S)-dependence in the first-order regret term. It would be interesting to design algorithms that require only convex optimizations but are still poly(S)-free. Thank you for the suggestion regarding non-convex optimization for poly(S)-free regret. We will clarify this in the introduction of our final version.
>
> **Regarding Question 2**: *Unbounded self-concordant GLMs*.
> Indeed, it does seem that our analysis extends to unbounded, self-concordant GLMs as well. However, we expect to incur additional log factors ($\log{T}$) in the regret. Taking Gaussian rewards as an example, the confidence interval would remain unchanged. We use the upper bound on the reward ($R$) in several places during the analysis (eg., in Lemma A.5) or in the algorithm (while defining $\beta(x)$) – similar arguments can be made for Gaussian rewards as well. The analysis for Gaussian rewards follows from the fact that Gaussian random variables remain bounded with high probability, allowing for extensions of our analysis.
> While such extensions are interesting, the current work focuses on the GLM model proposed in [8] and addresses the challenges in the limited adaptivity setting.
>
>
> **Regarding Question 3**: *Regarding l2-penalized MLE*.
> We have intentionally separated the convex optimization part and the (non-convex) projection step. This separation ensures that certain properties can be obtained before and after the projection; see, e.g., equation (26), which would not hold if we included the projection into the considered convex optimization problem. Moreover, it is not clear how the non-convex optimization projection can be included as a constraint in the log-loss optimization, while ensuring that the desired properties hold. Appendix E provides additional details in this direction.
>
> **Regarding Question 4**: *Geometry-dependent norm for warm-up*.
> We thank the reviewer for this helpful suggestion. We will include this in the updated version of the paper. Indeed, while the worst-case regret guarantee remains unchanged, we can use the geometry-dependent norm during warmup. Also, the algorithms analysis remains essentially the same.
>
>
> **Regarding Question 5**: *Ellipsoidal projection*.
> The reason why restricting optimization (6) to the ellipsoid around $\theta_o$ leads to tighter regret is because for the Non-”Switching Criterion I” rounds, it is guaranteed that $\lVert x \rVert_{V^{-1}}  \leq O(1/\sqrt{\kappa})$. Hence $\langle x, \theta^* - \theta_o \rangle \leq ||x||_{V^{-1}} || \theta^* - \theta_o ||_V \leq O(1/\sqrt{\kappa}) \gamma \sqrt{\kappa}$. Therefore, the main idea is in the design of the Switching Criterion I and not just in the ellipsoidal projection. Whether one can combine this Switching Criterion with existing algorithms and obtain better guarantees is an interesting direction worth exploring.
>
> **Regarding Question 6**: *Best arm identification*.
> This is an interesting question that complements the paper’s goals. It is, however, worth noting that the current warm-up conditions are designed to address changing contexts. Applying them to static arm sets may be suboptimal. For static arm sets, one can simply allocate initial rounds for warmup, as in [7]. The switching criteria-based warmup is needed in the current work because we are dealing with adversarial contexts. Overall, it remains unclear as to what one would gain by using the criteria for static arm sets.
>
> Note: All reference numbers are same as in the submitted paper.

---

> > ### Comment · Reviewer_bRFJ · 2024-08-11
> >
> > Thank you for the responses. They cleared up most of my concerns, and I'll be keeping my score and advocating for acceptance.
> >
> > One more follow-up question to Question 4:
> >
> > - so... using $H$-norm doesn't help with the analysis? I expected that as $H$ and $V$ may differ by a factor of $\kappa$, using $H$-based warmup would help significantly, e.g., [7] and Mason et al. (2022) ([7] in my above response). Can the authors elaborate on why using $H$ doesn't lead to any significant improvement? Moreover, *definitely not for the current rebuttal*, but I would be curious to see if there are any numerical differences in using $H$-based warmups.
> >
> > Lastly, one additional suggestion: please consider including a table of contents.

---

> > > ### Author Response · Authors · 2024-08-13
> > >
> > > Thank you for the relevant suggestions and for your support in advocating for acceptance. We will include a table of contents in the updated version.
> > >
> > > Regarding the use of the $H$-norm in warmup, the $\kappa$ factor in the lower-order term of the regret during the warm-up rounds arises from our approach of lower-bounding the matrix $H^* = \sum_t \dot{\mu}(x_t^T \theta^*)  x_t x_t^T $ by $H^* \succeq \frac{V}{\kappa}$, where $V = \sum_t x_t x_t^T$ (similar to Jun et al. 2021 or [7] for burn-in phase). Given $\lVert \theta^* \rVert \leq S$, one might consider a tighter bound, such as $H^* \succeq \sum_t \dot{\mu}(\lVert x_t \rVert S) x_t x_t^T$ (referred to as $H^\text{naive}$ in Mason et al. 2022). However, this bound still incurs a $\kappa$ factor in the worst-case for certain arm sets, e.g. when all arms have the same length ($\lVert x \rVert = 1 \, \forall x \in  \mathcal{X}$).
> > >
> > > That said, for arm sets where $\lVert x \rVert$ varies significantly, using $H^\text{naive}$ may improve empirical performance, making it a worthwhile direction for future exploration. We will incorporate this idea from Jun et al. 2021 and Mason et al. 2022 in the updated version.

---

### Official Review · Reviewer_wYhM · 2024-07-18

**Soundness:** 3
**Presentation:** 2
**Contribution:** 3
**Rating:** 7
**Confidence:** 3

**Summary:**

This paper considered regret minimization for a generalized linear reward model with limited adaptivity, in which the set of arms $\mathcal{X}t$ is stochastically generated by unknown distribution $\mathcal{D}$, and after pulling $x_t \in \mathcal{X}_t$ the learner receives a reward $r_t$ sampled from the GLM distribution $P(r|x_t)$ with unknown $\theta^*$.
In the first setting M1, the algorithm is given a budget $M$ and is asked to decide upfront $M$ rounds to update its policy. For M1, B-GlinCB is proposed, whose regret depends on $\sqrt{\hat{\kappa}^{-1}}$ or $\sqrt{{\kappa^*}^{-1}}$.  In the second setting M2, the algorithm is given a budget $M$ and needs to decide $M$ rounds to update its policy adaptively.
For RS-GlinCB, the authors provided regret bound where $\kappa$-dependence only appears in $\log^2T$ term. Experimental results are demonstrated to validate their algorithms.

**Strengths:**

The first attempt to study GKM with limited adaptivity. The algorithm removes the instance^dependent non-linearity parameter $\kappa$ which can be exponentially large concerning $|| \theta^*||$.

**Weaknesses:**

1. To compute the length of the warm-up batch in M1, the knowledge of $\kappa$ which depends on the optimal parameter $\theta^*$ is required. Also, UCB/LCB scores and $\beta(x)$ also need this knowledge. In M2 setting, Switching Criterion I depends on $\kappa$. In practice, how can we estimate $\kappa$ parameterized by unknown $\theta^*$?


2. The reviewer could not understand how significant the benefit of removing $\kappa$dependence compared with other $\sqrt{\hat{\kappa}^{-1}}$ or $\sqrt{{\kappa^*}^{-1}}$ is while allowing the policy to update at most $O(\kappa \log^2 T)$ times that is $\kappa$-dependent. Only when $\kappa =o(\log T)$, the amount of adaptivity is reasonable, but now the regret bound does not need to care about $\kappa$-dependence in such cases. Conversely, when $\kappa$ is large, removing such dependence from the regret bound is important but now the algorithm requires a large amount of updates.  Could authors discuss the benefits/trade-offs of this point more?

**Questions:**

Please see questions in Weaknesses.


Typo/minor comments:

$x^*$ is dependent on arm set $\mathcal{X}$, so other notation may be clearer such as $x^*_t$ for $\mathcal{X}_t$.

Since the def. of $\kappa$ in Alg 1 and Alg 2 is different, why not introduce different notations?

How the scaling parameter $\beta(x)$ is defined in (2)? Is this same value defined later in (3)?

**Limitations:**

Yes

---

> ### Author Rebuttal · Authors · 2024-08-07
>
> We thank the reviewer for the feedback. We address the comments and questions below:
>
> **Regarding Weakness 1**.
> *(1a) Unknown $\kappa$*.
> It is relevant to note that an upper bound on $\kappa$ suffices for the mentioned use cases. $\kappa$ is an instance-dependent parameter that captures the non-linearity and quantifies the hardness of the GLM instance.
> There is no cyclic dependency here between estimating $\theta^*$ and $\kappa$. In particular, it is standard in bandit literature (specifically, linear and generalized linear bandits) to assume an upper bound on $\theta^*$. Such an upper bound directly translates into an applicable upper bound for \kappa; see discussion below.
>
> *(1b) Dependence of $\kappa$ on* $\theta^*$.
> We note that the inclusion of  $\theta^*$ in the definition of $\kappa$,  $\left( \kappa = \max_{x \in \cup_{t=1}^T  {\cal X}t} \frac{1}{\dot{\mu} ( \langle x, \theta^* \rangle )}  \right)$ is beneficial. Prior works, such as [1],[2] use the following definition $$\kappa = \max_{\lVert \theta \rVert \leq S } \max_{x \in \cup_{t=1}^T  {\cal X}t} \frac{1}{\dot{\mu} (\langle x, \theta \rangle)} $$ which also appears in their regret expression. Our definition is much tighter, leading to potentially smaller regret.
>
>
> **Regarding Weakness 2**.
> Here, it is best to not conflate $\kappa$ and $T$. $\kappa$ is an instance dependent parameter, which is potentially large, but fixed for the instance. The number of rounds, $T$, on the other hand, is a growing quantity.
>
>
>
> **Regarding the definition of $\beta(x) and minor comments**.
> Yes, the understanding is correct. We will move this definition next to the first expression in the final version.
> Thank you for pointing out the minor typos. We will fix these in the updated version of the paper.
>
> [1] Filippi, Sarah, et al. "Parametric bandits: The generalized linear case." Advances in neural information processing systems 23 (2010).
> [2] Faury, Louis, et al. "Jointly efficient and optimal algorithms for logistic bandits." International Conference on Artificial Intelligence and Statistics. PMLR, 2022.

---

> ### Comment · Reviewer_wYhM · 2024-08-07
> **Thank you for your response**
>
> Dear Authors,
>
> Thank you for your response, in which the following points have been resolved:
> - (1) In the algorithm design, the access to an upper bound is sufficient.
> - (2) Other questions such as the definition of $\kappa$ and its difference from prior work.
> - (3) The necessary assumptions in the paper are standard in the field of GLMs.
>
> I found that the analysis in this paper is very well-developed, particularly the self-concordance of bounded GLMs. Therefore, I have changed my score to Accept.

---

> > ### Author Response · Authors · 2024-08-11
> >
> > We thank the reviewer for updating the score.

---

### Decision · Program_Chairs · 2024-09-25

**Decision:**

Accept (spotlight)

**Comment:**

The paper proposes novel algorithms for the contextual bandit problem with bounded glm rewards when the number of updates is limited. An algorithm which determines in advance the rounds at which the policy will be updated is proposed for the case where the contexts follow a stochastic distribution. Another algorithm which determines the udpate rounds in an adaptive manner is proposed for the case where the contexts can possibly be adversarial. Both algorithms are proven to achieve $\tilde{O}(\sqrt{T})$ regret with only $\Omega(loglogT)$ and $\tilde{O}(log^2T)$ updates respectively under the respective settings. As an additional contribution to the glm bandit literature, the authors manage to remove the $\kappa$ dependency in the regret bound, where $\kappa$ is an model dependent parameter that captures non-linearity of the reward model. This is achieved by a novel design component of the algorithm; utilizing a design matrix appropriately scaled via $\kappa$.

The paper has numerous contributions to the glm bandit literature (achieving kappa independent regret for glms beyond logistic models, removing polynomial dependency on the norm of the parameter), and the proposed algorithms are of great practical use as they achieve tight regret bounds with limited number of udpate rounds.

One weakness pointed out by multiple reviewers was that the knowledge of kappa is necessary to run the algorithm. Authors adequately addressed this point in their rebuttal by explaining that the upper bound of kappa is sufficient, and that the assumption that this upper bound is known is a standard assumption in the glm bandit literature.

Authors are stronlgy encouraged to incorporated suggestions of the reviewers in their final version of the paper

- They should add some baselines proposed by Reviewer bRFJ in their experiments
- Some missing reference on logistic bandits should be added
- It should be stated that the poly(S)-independence relies on solving a nonconvex optimization problem